


# Daytime low-level clouds in West Africa – occurrence, associated drivers and shortwave radiation attenuation

Derrick K. Danso[1,2], Sandrine Anquetin[1], Arona Diedhiou[1,2], Kouakou Kouadio[2], Arsène T. Kobéa[2]

[1]Université Grenoble Alpes, IRD, CNRS, Grenoble-INP, IGE, 38000 Grenoble, France.

[2]Laboratoire de Physique de l'Atmosphère et de Mécaniques des Fluides (LAPAMF), Université Félix Houphoüet, Boigny, Abidjan, Côte d'Ivoire.

*Correspondence to*: Derrick K. Danso (derrick.danso@univ-grenoble-alpes.fr)

**Abstract.** This study focuses on low-level clouds (LLC) that occur during the daytime over West Africa (WA). These daytime LLCs play a major role in the earth's radiative balance, yet, their understanding is still relatively low in WA. We use the state-
of-the-art ERA5 dataset to understand their occurrence and associated drivers as well as their impact on the incoming surface solar radiation in the two contrasting Guinean and Sahelian regions of WA. The diurnal cycle of the daytime occurrence of three LLC classes namely No LCC, LLC Class-1 (LLCs with lower fraction), and LLC Class-2 (LLCs with higher fraction) are first studied. The monthly evolutions of hourly and long-lasting LLC (for at least 6 consecutive hours) events are then analyzed as well as the synoptic and local dynamics associated with the long-lasting LLC events. The occurrence of No LLC
events does not present any specific correlation with the time of the day, whatever the region while as soon as LLC coverage becomes more pronounced (LLC Class-2), a diurnal evolution is noted and appears to be strongly different from one region to the other. During the summer months in the Guinean region, the occurrence of LLC Class-1 is low while LLC Class-2 is frequent (occurrence frequency around 75 % in August). In the Sahel, LLC Class-1 is dominant in the summer months (occurrence frequency more than 80 % from June to October), however the peak occurrence frequency of Class-2 is also in
the summer. In both regions, the occurrence of LLCs during the rainy season is associated with an influx of cold moist air driven by strong southwesterly winds from the Guinean Gulf. Furthermore, the occurrence of LLC Class-2 is linked to strong surface heating and evaporation of soil moisture. During the dry season on the hand, the occurrence of LLCs is linked more to turbulent upward motion of air caused by surface heating (only in the Sahel) and the convergence of air masses near the surface. The results also showed that the occurrence of LLC Class-2 causes high attenuation of the incoming solar radiation, especially
during JAS where about 49 % and 44 % of the downwelling surface shortwave radiation is lost on average in Guinea and Sahel respectively.



# 1 Introduction

In West Africa (WA), the prediction of key features of the climate such as the West African Monsoon (WAM) is known to
have large uncertainties (Christensen et al., 2013). Most of the uncertainties are essentially linked to the inability of climate
models to efficiently represent convection and, in particular low-level clouds (LLCs) (Hannak et al., 2017). Climate models
struggle to properly reproduce the planetary boundary layer above the ocean and the continental surface as well as the land-
ocean flux gradients that drive the triggering and the variability of LLC systems. These clouds and their temporal variation
strongly influence the reflectivity of the atmosphere. As a result, large uncertainties exist in the prediction of surface solar
radiation in the region (Knippertz et al., 2011). In the light of the recent escalation of interest in solar energy projects in WA
(World Bank, 2019) after the Paris Agreement in 2015, predictability of surface solar radiation (Armstrong and Hurley, 2010;
Kosmopoulos et al., 2015) thus, remain a key issue for any feasibility study concerning solar energy during either dry or wet
seasons in the region.

WA is characterized by a frequent occurrence of different cloud types. Near the Gulf of Guinea in the southern part of WA,
LLCs are prevalent all year round (Adler et al., 2017; Danso et al., 2019; van de Linden et al., 2015). Further north in the
Sahelian region, these clouds are frequently observed during the WAM season from July to September (Bouniol et al., 2012).
These LLCs consist of stratified clouds, most of which are nocturnal low stratus clouds covering wide areas and persisting into
the early afternoons (Babić et al., 2019a; Schuster et al., 2013) to shallow convective clouds. LLCs composed of liquid water
are well known to have a large impact on radiative transfer (Hill et al., 2018; Liou, 2002; Turner et al., 2007). As a result, they
drive the diurnal cycle of convection and the planetary boundary layer (Gounou et al., 2012; Grabowski et al., 2006). During
the daytime, these clouds block a large portion of the incoming shortwave radiation reducing the irradiance at the earth's
surface.

Several studies have been carried out in WA to understand LLCs, their variability, and the processes that aid in their formation,
maintenance, and dissipation (e.g., Dione et al., 2019; Kalthoff et al., 2018; Schrage and Fink, 2012; Schuster et al., 2013).
Majority of those studies were carried out within the framework of two major field campaigns; the African Monsoon
Multidisciplinary Analysis (AMMA) (Redelsperger et al., 2006) in 2006 and the Dynamics–Aerosol–Chemistry–Cloud
Interactions in West Africa (DACCIWA) (Knippertz et al., 2015) in 2016. Both of these field campaigns provided an
unprecedented database of surface-based observations of different cloud properties. Other studies on LLCs in the region have
also been performed with satellite observations and model data (Adler et al., 2017; Hannak et al., 2017; van de Linden et al.,
2015). These studies suggest that LLCs formation is linked to a number of processes including but not limited to cooling
caused by horizontal cold air advection and the occurrence and strengthening of the nocturnal low-level jet (Adler et al. 2017;
Babić et al. 2019b; Schrage and Fink 2012).

Most of these studies, however, focused on night-time LLCs in the southern part of WA (e.g., Adler et al., 2019, 2017; Babić
et al., 2019b; Schrage and Fink, 2012; Schuster et al., 2013). These nocturnal LLCs do not have any direct impact on the
surface solar irradiance during the daytime but their presence could provide an idea of the cloudiness the next day (at least





early in the morning). Consequently, the conditions associated with LLC occurrence during the daytime in the region are less documented. As a result, our understanding of the impacts of these clouds on surface solar irradiance during the daytime is relatively limited. Moreover, most of the previous studies have been limited the WAM season (e.g., Dione et al., 2019; Knippertz et al., 2011; van de Linden et al., 2015; Schuster et al., 2013). This is understandable as WA is cloudiest during the

WAM season. However, it was shown that daytime LLCs exist in all seasons especially over southern WA (Danso et al., 2019). It is therefore important to understand the nature of such daytime LLCs over the whole region and not only during the WAM season. This is particularly challenging due to the lack of a long-term surface-based observational dataset in WA, limiting our understanding of these cloud systems. Few studies which were done with simulations, reanalysis, and satellite data in the region (e.g., Adler et al., 2017; Knippertz et al., 2011; van de Linden et al., 2015; Schuster et al., 2013) have nevertheless

shown results similar to those of ground observational studies to some extent.

In this general context, the main contribution of this study is to enhance our understanding of daytime LLCs in WA by focusing on daytime LLCs in both the dry and wet seasons with a state-of-the-art reanalysis dataset. Firstly, we will present the seasonal and diurnal distributions of the occurrence frequency of the daytime LLCs in two areas of WA with contrasting climate regimes (the Sahelian and the Guinean regions). We will then identify synoptic- and local-scale conditions associated with the presence

of the daytime LLCs. Additionally, we will investigate the percentage of incoming surface shortwave (SW) radiation that is attenuated when these clouds are present.

This paper is organized as follows: Sect. 2 describes the study area, datasets, and methodology. Section 3 presents the diurnal and monthly distribution of occurrence of daytime LLCs while the synoptic and local conditions associated with the occurrence of long-lasting LLCs are discussed in Sect. 4. Section 5 presents the estimated attenuation of incoming shortwave radiation

during the occurrence of LLCs. Finally, conclusions are drawn in Sect. 6.

## 2 Data and Methods

### 2.1 Data

In this study, we analyze the 5[th] generation of the European Centre for Medium-range Weather Forecasts (ECMWF) Re-analysis – ERA5[1] (Copernicus Climate Change Service, 2017) to understand the occurrence of LLCs and their associated

synoptic conditions and land surface fluxes and to discuss their impacts on incoming shortwave radiation in WA (Fig. 1). The reanalysis is available at a temporal resolution of an hour and has a native horizontal resolution of 0.25°. However, due to more emphasis on large-scale conditions in this study, the data was directly extracted on a regular 1° x 1° grid from ECMWF. A period of 10 years from 2006 to 2015 was analyzed for the study. For this period a total of 43800 data points consisting of only daytime hours from 0600 h to 1700 h, were thus considered.

---

[1] Detailed documentation at https://confluence.ecmwf.int/display/CKB/ERA5+data+documentation





ERA5 provides different variables of cloud cover properties. These cloud variables are produced in CY41R2 of the ECMWF's Integrated Forecasting System and documented in Part IV of the IFS Documentation[2]. The cloud cover variables consist of cloud fraction and cloud hydrometeors (liquid or ice water) and are available as 3D or 2D distributions. Here, we use the 3D distribution of cloud liquid and ice water content to show the atmospheric vertical profiles of clouds in selected areas over WA. We then use the 2D single-level reanalysis of cloud fraction to define the occurrence of LLCs. The atmospheric conditions

and radiative impacts are analyzed based on this occurrence. In a recent study by Danso et al., (2019), the ERA5 cloud fraction was evaluated against some available synoptic in situ observations of the Integrated Surface Database (Lott et al., 2001). This assessment revealed a reasonable agreement between ERA5 and the in situ cloud fraction dataset (see Appendix S1 in Danso et al. (2019)). In this study, we focus on low clouds defined as all clouds integrated from the surface to 2 km altitude in the atmosphere (≈ 800hPa). In the ERA5 dataset, this includes all clouds with bases below 2 km.

Other ERA5 synoptic variables are analyzed to show some of the atmospheric conditions during the occurrence of LLCs and their surface heat fluxes explore to understand the potential interactions between the surface and the lower levels of the atmosphere. These include specific humidity, zonal and meridional wind components, vertical velocity, and the surface sensible and latent heat fluxes. Furthermore, we use the incoming downwelling shortwave radiation variable in ERA5 to estimate the percentage of incoming solar radiation that is attenuated in the presence of LLCs.

## 2.2 Methods

### 2.2.1 Identification of LCC occurrence

Our analysis is based on the occurrence of LLCs in two sub-regional windows over WA (rectangular boxes in Fig. 1) denoted here as Guinea (10° W to 0° E and 6° N to 9° N) and Sahel (10° W to 0° E and 12° N to 15° N). These areas are chosen primarily due to their contrasting climate regimes (Gbobaniyi et al., 2014). The Guinea region which includes countries like

Ghana and Cote d'Ivoire is currently undergoing significant economic and infrastructural developments and thus the need for clean energy keeps increasing. Consequently, there are many ongoing solar energy projects in this region such as the 155 MW Nzema solar PV power project in Ghana (Kuwonu, 2016), which when completed will be the largest solar farm on the African continent. Côte d'Ivoire has planned to increase its share of variable renewable energy up to 11 % by the end of 2020 and up to 16 % by 2030. Thus, a 25 MW solar power plant is under construction in the Northern part of the country in Korhogo while

two solar power plants projects with a capacity of 30 MW each have been approved and will be built in the localities of Tuoba and Laboa in the northwestern part of the country (MPEER, 2019). Additionally, there are several planned and existing solar energy projects in the Sahel window, especially in Burkina Faso with the 33 MW-capacity Zagtouli solar power plant (Moner-Girona et al., 2017) which is currently the largest solar energy project in the Sahelian part of WA.

Occurrence and no-occurrence of LLC are first identified from the extracted data points based on the quantitative value of the

cloud fraction (CF) in these two selected windows. This is used to show the diurnal and monthly distribution of the LLC

---

[2] Available at https://www.ecmwf.int/en/elibrary/16648-part-iv-physical-processes





occurrence. All timestamps with zero CF in the lower atmospheric level (up to 2 km) are designated as No LLC. Thus, hourly events of LLC occurrence are all timestamps with non-zero CF values including those with very small CF values. LLC events with very low CF values may show no or weak signals of the conditions that exist during their occurrence. Moreover, cloud radiative forcing is strongly dependent on the fractional coverage of clouds (Liu et al., 2011). Each LLC occurrence event is

therefore classified as one of two classes depending on the CF value for that event. The definitions of the classes are provided in Table 1.

Though the focus of this study is on LLC occurrence and the definition of the different classes is based on the CF in the lower atmosphere only, the convoluted cloud climatology in WA with frequent multilayer clouds (Stein et al., 2011) may also mean that other clouds at higher altitudes may exist in addition to LLCs. Fig. 2 illustrates this multilayer nature of clouds in WA in

connection with LLC occurrence, by showing the mean (of all hourly events) vertical profile of cloud water content of each class in the two selected regions. For LLC Class-1 and Class-2 in both windows, the trimodal vertical distribution of tropical convection (Johnson et al., 1999) indicates the presence of higher altitude clouds in addition to the LLCs. In the case of No LLC, high- and mid-level clouds could exist but with very low water content. This suggests and agrees with the satellite-based climatology of Stein et al., (2011) that LLCs frequently exist together with upper-level clouds.

**2.2.2 Identification of synoptic conditions during LLC occurrence**

Based on the different classes defined in Table 1, the regional atmospheric circulation and moisture flux were analyzed for the 10-year period of this study. For that, a composite approach was used to produce robust results based on a large number of events. We only take into account the occurrence of the different LLC classes that last for a sufficiently long time for their associated mean atmospheric and surface conditions to be significant at the regional scale. Therefore, an event here refers to

the occurrence of a given LLC class that lasts for at least six consecutive hours during the daytime. Composite analysis of the horizontal moisture flux advection ($Q_{adv}$) and wind at 950 hPa was performed. Here, $Q_{adv}$ is defined as:

$$Q_{adv} = \frac{1}{N}\sum_{i=i}^{N} q_i v_i \tag{1}$$

where $q$ is the specific humidity (in $g \cdot kg^{-1}$), $v$ is the meridional wind component (in $m\,s^{-1}$) and $N$ is the total number of events in a given LLC occurrence class. Since $q$ is always positive, $Q_{adv}$ can either be negative or positive depending on the

wind direction, with positive or negative values indicating northward and southward air movements respectively. Over the sea, positive values of $Q_{adv}$ from the ocean toward the continent thus suggest that moisture is likely transported onto the land along with the wind. In addition, the vertical velocity, horizontal air divergence, and anomalies of $q$ are analyzed together with $Q_{adv}$. Surface heat fluxes (sensible and latent heat) are known to have important contributions to the occurrence of LLCs (Ek and Holtslag, 2004). Air masses near the land surface can be lifted upwards by turbulent motions due to strong surface sensible

heat fluxes. On the other hand, soil moisture can be transported upwards through the effects of turbulent air motion by strong latent heat fluxes at the land surface. Here, we examined the local surface sensible and latent heat fluxes in the selected windows to understand the daytime interaction of the land surface with the boundary layer and LLC occurrence. Additionally,





the Bowen ratio ($BR$) (Lewis, 1995), the ratio between the sensible heat flux and the latent heat flux, was used to investigate the energy transfer between the land surface and the atmosphere in the two selected windows. The analysis was performed for

two seasons representing two contrasting climate regimes associated with the southernmost position of the Inter-Tropical Convergence Zone (ITCZ) over Guinea (in July – August – September (JAS)) and the northernmost position of the ITCZ over Sahel (in December – January – February (DJF)).

### 2.2.3. Determination of cloud shortwave attenuation effects

To investigate the quantity of incoming downwelling shortwave radiation attenuated in the presence of LLCs (in $W\ m^{-2}$) at a

given time $t$, the cloud shortwave radiative effect ($CRE_{SW\downarrow}$) was estimated by finding the difference between all-sky and clear-sky incoming shortwave radiation following Bouniol et al., (2012) and expressed with the simple equation:

$$CRE_{SW\downarrow}(t) = SW_{CS}^{\downarrow}(t) - SW^{\downarrow}(t) \qquad (2)$$

where, $SW^{\downarrow}$ is the downwelling shortwave radiation in all-sky conditions and $SW_{CS}^{\downarrow}$ is the theoretical value of the downwelling shortwave radiation in clear-sky conditions. It must be noted that $CRE_{SW\downarrow}$ is estimated for a given class of LLC occurrence.

To eliminate the influence of the wide range of daytime hours on its mean value, we expressed $CRE_{SW\downarrow}$ as a percentage of the attenuated $SW_{CS}^{\downarrow}$. Thus,

$$CRE_{SW\downarrow}(t) = \frac{SW_{CS}^{\downarrow}(t) - SW^{\downarrow}(t)}{SW_{CS}^{\downarrow}(t)} \times 100 \qquad (3)$$

### 3. Temporal distribution of occurrence of daytime LLCs

The daytime distribution of occurrence frequency of the three LLC classes are presented in Fig. 3 for the two selected areas.

In Guinea, the occurrence frequency of No LLC events does not present a very well-marked diurnal cycle with a slightly higher number of events occurring at 1700 UTC, though these events are extremely rare (less than 25 for the 10 years). In contrast, both LCC Class-1 and Class-2 occurrences present well-defined diurnal cycles. The occurrence of LLC Class-1 events is low during the early morning (lowest at 0700 UTC) and increases progressively throughout the day to late in the afternoon. On the contrary, LLC Class-2 events are high early in the morning (highest at 0700 UTC) but decrease continuously over the course

of the day to the late afternoon (1700 UTC). The early morning peak in the LLC Class-2 events could be as a result of the contribution of residuals from nocturnal low-level stratus clouds which persist long into the day (Kalthoff et al., 2018). These clouds usually have large fractional coverage during their occurrence in the night (van de Linden et al., 2015) but start to dissipate during the daytime (Kalthoff et al., 2018). The continuous decreasing and increasing in the event number of LLC Class-2 and LLC Class-1 respectively, from morning to late afternoon could, therefore, be linked to the gradual dissipation of

those nocturnal low-level stratus clouds during the day. Additionally, the early morning peak in the events of LLC Class-2 could also be partly linked to contributions from tropical oceanic low-level convection which is maximum during the early



morning (Yang and Slingo, 2001). The proximity of the Guinean region to the ocean means that convected air over the Gulf of Guinea could be transported inland which will, in turn, enhance LLC formation.

In the Sahel region, events of No LLC occur without a well-marked diurnal cycle but are slightly lower around midday.
Similarly, LLC Class-1 events do not present a very distinct diurnal cycle. On the other hand, events of LLC Class-2 show a clear and well-marked diurnal cycle with a higher occurrence frequency in the late morning to midday (highest at 1100 UTC) and lower values during early mornings and late afternoon. This tendency seems to be strongly controlled by surface processes such as continental daytime solar heating (Mallet et al., 2009). Moreover, it could further be related to the occurrence of deep mesoscale convective systems (MCSs) which present a fairly similar evolution (both seasonally (Fig. 4) and diurnally) in this
region of WA (Vizy and Cook, 2018). It is, therefore, reasonable to assume that most of these LLC Class-2 events in the Sahel are the well documented deep MCSs which are responsible for around 90 % of total rainfall in the region (Goyens et al., 2012; Mathon et al., 2002a; Vizy and Cook, 2019).

The monthly distribution of the occurrence frequency (hourly) of the three LLC classes is presented in Fig. 4 for the two selected regions (occurrences that last for at least six consecutive hours are in shown as black bars). In Guinea, events of No
LLC are very rare (less than 200 in the 10 years data) and occur only in the DJF. On the other hand, LLCs occur in all months in this area. The occurrence of LLC events with lower fractional coverage (LLC Class-1) shows a clear seasonal cycle with the highest number of events in the dry and transitional seasons and the lowest number of events during JJAS which corresponds to the core of the monsoon season (Sultan et al., 2003). Coincidentally, LLC events with higher fractional coverage (LLC Class-2) occur most during the monsoon season. This period is associated with moisture influx from the Gulf of Guinea
(Sultan and Janicot, 2003; Thorncroft et al., 2011) which enhances low clouds formation over the region during this period. Interestingly, the peak of LLC Class-2 events occurs in August, the so-called "little dry season" in southern WA (Adejuwon and Odekunle, 2006; Chineke et al., 2010; Froidurot and Diedhiou, 2017). Whether this peak of LLC Class-2 events in August which coincides with reduced rainfall in southern WA is due to an increased occurrence of non-rain-bearing low clouds (and a decrease in rain-bearing ones) or not is beyond the scope of this paper but needs to be investigated in future.

In Sahel, events of No LLC are rather frequent (unlike Guinea) and occurs in every month except in August when the monsoon is fully developed over the Sahelian region of WA (Sultan et al., 2003; Thorncroft et al., 2011). Similarly, LLC Class-1 events are frequent throughout the year but with the highest occurrences from July to October (with a slight decrease in August) and lowest occurrences during the dry season. LLC Class-2 events are marked with a strong seasonal signature, occurring only from July to October with a distinct peak occurrence in August when the Inter-Tropical Convergence Zone (ITCZ) is at its
northernmost latitude over WA (Nicholson, 2009, 2018; Sultan and Janicot, 2000). This period also coincides with the maximum rainfall over the Sahelian region (Nicholson, 2013) and suggests a strong link between the occurrence of LLC Class-2 events and the Sahelian rainfall. Again, Mathon et al., (2002b) showed that cloud cover associated with a sub-population of MCSs in the Sahel is modulated at the synoptic-scale during the African easterly wave activity, with an increase of the cloud cover in and ahead of the trough and maximum rainfall behind of the wave trough.






## 4. Synoptic conditions related to the occurrence of daytime LLCs

The synoptic conditions associated with the occurrence of the different LLC classes are mainly presented for the JAS and DJF seasons. Only occurrences that last for at least six consecutive hours are considered to analyze these conditions (i.e., here, an event refers to an occurrence that persists for at least six consecutive hours). Table 2 presents the number of the 'six-consecutive

hours' events extracted for each class and for both seasons over the 10 years. During the JAS season, no occurrence of No LLC events was detected in both Guinea and Sahel window (i.e., daytime LLCs are always present in both regions during the JAS). LLC Class-1 events dominated the Sahel region during JAS (with 824 events) but on the contrary, LLC Class-2 events dominated the Guinea region (with 812 events). In DJF, No LLC events dominated the Sahel (with 876 events) and no LLC Class-2 occurrence has been identified in this area during this season. In Guinea, on the other hand, LLC Class-1 occurrences

dominated the area during DJF (with 888 events). Composites of the different variables are thus analyzed and then discussed only based on the extracted events for each class and each season, as presented in Table 2.

### 4.1. Atmospheric circulation and moisture advection

The occurrence of daytime LLC events in the selected windows is related to large-scale dynamics of the atmosphere over the WA region. Fig. 5 and 6 present the composites of $Q_{adv}$ and the atmospheric circulation at 950hPa for JAS and DJF

respectively, and for the two studied regions. During JAS in both regions (Fig. 5), the occurrence of LLC Class-1 and Class-2 events is related to an influx of cold moist air from the ocean associated with the strong southwesterly winds (positive $Q_{adv}$) from the Guinean Gulf into the continent. Predictably, the horizontal advection of moist air is stronger during the occurrence of LLC Class-2 events than LLC Class-1 events. This is true for LLC occurrence in both regions however, the advection is more intense for LLC occurrence in Sahel in terms of the inland penetration of the cold moist. This inland advection of moist

air from the ocean has been found to play a major role in cooling (Adler et al., 2019b) which in turn enhances saturation of water vapour and consequently LLC formation (Adler et al., 2019, 2017; Babić et al., 2019b).

During DJF in both regions (Fig. 6), events with No LLC are characterized by dry northeasterly winds that inhibit LLC formation. The occurrence of LLC Class-1 events in Guinea and Sahel is characterized by southerly winds just at the southern coast but do not spread into the area of the windows. Other factors are likely responsible for the formation of these events in

both regions during DJF. Nevertheless, the LLC Class-1 events during DJF (in Sahel) are related to positive $q$ anomalies that seem to have been transported onto the continent from the North Atlantic (see Fig. A1a in Appendix) through a modulation by the West African Heat Low (Lavaysse et al., 2009) but this needs to be investigated further. LLC Class-2 events in Guinea are associated with a much stronger moisture advection into the region (Fig. 6) as well as a positive anomaly of $q$ within and around the window (Fig. A1b in Appendix).

Unlike in JAS, the lack of moist air advection from the Guinean Coast during LLC occurrence in DJF (at least for LLC Class-1 events) in both regions suggests that other factors are at play. Fig. 7 shows the mean vertical profiles of horizontal air divergence and the vertical velocity in the two regions for the first 2 km of the atmosphere during DJF. The horizontal





divergence is used to quantify the rate at which air is spreading out of a point per unit area, while vertical velocity measures the speed of air motion in the upward or downward direction. In keeping with the ECMWF convention, negative values for
both quantities will be favorable for cloud formation i.e., converging air has negative divergence, and ascending air has negative vertical velocity. In both regions, the occurrence of LLC Class-1 events is associated with the convergence of air within the first 1km of the atmosphere (negative mean divergence, Fig. 7). Additionally, the occurrence of LLC Class-2 events in Guinea is also associated with converging air. Similarly, the vertical motion of the air is in the upwards direction on average (negative vertical velocity) in both regions during LLC occurrence. The upward air motions are likely caused by the
convergence of air masses near the surface as shown in Fig. 7. It is worth noting that on average during No LLC events in Guinea, there is a weak upward motion associated with a maximum convergence around 500 m and a maximum divergence at 1500m when in the Sahel, the vertical velocity is on average neutral while the maximum convergence at the surface is very weak and close to the surface (250 m) and the maximum divergence is reached at 750 m. It is also important to note that during JAS, the average vertical profiles of these processes (divergence and vertical velocity) are not similar to DJF (not shown).

**4.2. Surface heat fluxes**

Fig. 8 presents the monthly mean anomalies of both surface sensible and latent heat fluxes computed over the two selected regions for the three LLC occurrence classes. For the No LLC events class, in both regions, a land surface cooling is present throughout the year as shown by the negative anomalies of the sensible heat. This surface cooling thus reduces upward motions of the air masses near the surface and contributes to inhibiting cloud formation regardless of the positive anomalies of surface
latent heat flux (at least from April to June over Sahel). The lack of upward motion of air in both regions during No LLC events is also seen in the vertical velocity profiles in Fig. 7 (almost neutral mean vertical velocity). During the summer months there can be moisture influx extending into Sahel region but the surface cooling (negative anomaly of sensible heat) reduces the upward rise of air near the surface.

In Sahel, the occurrence of LLC Class-1 events in all months except in JAS is associated with surface warming (positive
anomalies of surface sensible heat, up to about 25 W m$^{-2}$ in January). This surface warming could explain the upward air motion in DJF (Fig. 7) that could lead to the LLC formation. During JAS, these events occur with negative anomalies of sensible heat (about -3 W m$^{-2}$ in August when cloud coverage in Sahel is strongest (Fig. 4)). It is possibly due to relatively intense cloud coverage which reduces incoming shortwave radiation to warm the surface. Similarly, negative anomalies of the surface latent heat flux are associated with these events in the Sahel during most of the year (except in November and December
even though these anomalies are rather small). This may indicate that the occurrence of LLC Class-1 events during the JAS season in the Sahel is more linked to the large-scale factors (such as $Q_{adv}$ as shown in Fig. 5) rather than local factors such as the surface heat fluxes. However, during the DJF season, the surface sensible heat flux (but not the surface latent heat) and possibly other large-scale processes mainly drive the occurrence of LLC Class-1 events in Sahel. Moreover, the surface sensible heat flux seems to play a more important role in the occurrence of LLC Class-1 events in Sahel (except in JAS) than
in Guinea. In the latter region, both surface sensible and latent heat fluxes do not seem to play a crucial role in the occurrence





of the LLC Class-1 events, since they present negative anomalies during the occurrence of these events with the lowest values during the core of the monsoon season when cloud coverage is much intense in the region.

On the contrary, the occurrence of LLC Class-2 events is more linked to the surface heat fluxes in both regions. Both regions show positive anomalies for both surface sensible heat flux and the latent heat flux indicating surface warming and soil
moisture evaporation. This suggests that the occurrence of these types of LLC depends on both regional-scale dynamic features and local-scale features. Surface sensible and latent heat fluxes combined with low-level convergence (Fig. 7) lead to turbulent upward motion of water vapour which is further cooled by moist advected air from the Guinean Coast, leading to saturation and then LLC formation. The role of the surface heat fluxes though appears to be much critical in Sahel than in Guinea. This is shown by the very large anomalies of surface sensible and latent heat fluxes in Sahel compared to Guinea (sensible heat
anomaly of about 45 W m$^{-2}$ against 5 W m$^{-2}$ in July and latent heat anomaly of about 40 W m$^{-2}$ against 5 W m$^{-2}$ in in October). Furthermore, surface sensible heat flux seems to have a large influence on the LLC occurrence than the surface latent flux at the beginning of JAS in the Sahel. This outcome is in agreement with Couvreux et al., (2012) who found that in this semi-arid region of WA, the existence of a large surface sensible heat flux is responsible for the occurrence of deep convective systems (as already mentioned in Sect. 3 most of these Class-2 events in Sahel are clouds with deep vertical extents). In other parts of
the world (e.g., Bosman et al., 2019 and Tang, 2004), it was shown that the surface sensible heat flux has a much significant role in LLC formation than the surface latent heat flux.

In addition to the surface sensible and latent heat flux anomalies, the monthly mean BR computed over the two selected regions associated with the three LLC occurrence classes is presented in Fig. 9. For the occurrence of both LCC Class-1 and Class-2 events, BR values are much larger before and after the monsoon season. This is very clear especially in the Sahel (for LLC
Class-1 occurrence) and is a result of large sensible heat fluxes and low latent heat fluxes at the surface due to lack of moisture in the soil during this time of the year (Ramier et al., 2009). This means that most of the energy for heat transfer near the surface goes into heating the ground (and the air in contact with it) rather than evaporating ground moisture. It further confirms our assertion that sensible heat (and large-scale features) plays a much important role in the occurrence of LLC Class-1 events (mostly during the dry season in this region) than latent heat. However, during the monsoon season, the occurrence of both
LLC Class-1 and LLC Class-2 events is characterized by low values of BR. Soil moisture becomes a particularly important variable to initiate deep convections which leads to the occurrence of LLC (especially Class-2), as indicated by the low values of BR. This is in agreement with the findings of Alonge et al., (2007) over the Sahel region.

## 5. Attenuation of incoming shortwave radiation during LLC occurrence

In Fig. 10 and 11, the regional distribution of cloud radiative effect on incoming downwelling shortwave radiation during JAS
and DJF is presented respectively as the percentage of incoming shortwave radiation attenuated during the occurrence of LLCs in the two regions as computed with Eq. (3) ($CRE_{SW\downarrow}$). In this section, we will focus more on the mean $CRE_{SW\downarrow}$ in each of the regions. Firstly, there is a strong seasonal variability of $CRE_{SW\downarrow}$ in the region. It is obvious that shortwave attenuation is much



higher during JAS than during DJF over the entire region as a result of increased cloud coverage during the monsoon season (Bouniol et al., 2012; Danso et al., 2019). This is true whether LLCs are present or not. During the occurrence of LLCs, the

values of $CRE_{SW\downarrow}$ are much lower for LLC Class-1 than LLC Class-2 events, whatever the region and the season. This clearly shows the strong dependence of $CRE_{SW\downarrow}$ on the fractional coverage of clouds as previously discussed by (Liu et al., 2011).

In JAS, the strong impacts of LLCs on incoming shortwave radiation is seen from the high $CRE_{SW\downarrow}$ values during LLC occurrence. During the occurrence of LLC Class-1 events, the mean percentage attenuation of incoming shortwave radiation becomes 29.9 % and 17.5 % respectively in the Guinean and Sahelian regions. In the same way, 49.1 % and 44.2 % of incoming

shortwave radiation are attenuated on average in Guinea and Sahel, respectively during events of LLC Class-2. Though these values are attenuated in the two regions under consideration, higher losses can be experienced in different areas over the region during those events. For instance, during LLC Class-2 events in the Sahel, the areas around the southern coast and the Guinea highlands experience much higher $CRE_{SW\downarrow}$ values (up to 65 %). This could be due to the presence of more clouds in those areas which extend into the Sahel region.

During DJF, during events of No LLC in Guinea and Sahel, about 2.2 % and 2.3 % (Fig. 11) of the incoming shortwave radiation is lost on average respectively. Even though there is no LLC occurrence during these events, this attenuation value is due to the possible presence of other cloud types at the upper levels. However, similar to the JAS season, the occurrence of LLCs leads to higher $CRE_{SW\downarrow}$ in both regions. In Guinea, 13.3 % of incoming shortwave radiation is lost on average during the occurrence of LLC Class-1 events while 32 % is lost during LLC Class-2 events. In Sahel, the mean attenuation during the

occurrence of LLC Class-1 events is 16.3 % though areas around the southern coast of WA can experience higher losses.

The large attenuation of incoming shortwave radiation during LLC occurrence (especially LLC Class-2) could be detrimental to planned solar energy application projects if not properly planned. For instance, during the JAS season in the Guinea region (and southern part of WA in general), the dominance of LLC Class-2 events (Table 2 and Fig. 3) means that surface solar radiation will be significantly reduced for a large number of periods. Although there are equally high losses of incoming solar

radiation in the Sahel during LLC Class-2 events in JAS, these events are much lower than in the Guinea region. Besides, the Sahel region has several persistent periods with no LLCs compared to the Guinea region in the other months of the year, especially during DJF. However, other issues that are not discussed in this paper such as dust (e.g., Bonkaney et al., 2017) could severely hinder solar energy projects in the Sahel region. In all cases, the implementation of large-scale solar energy projects in both regions requires meticulous planning to ensure that the intense cloudiness over the region and the high

variability of these clouds as shown by Danso et al., (2019) do not compromise the long-term aims of those developments.

## 6. Conclusion

In this paper, we made use of the state-of-the-art hourly reanalysis dataset from ECMWF – ERA5 from 2006 to 2015 to analyze the occurrence of daytime LLCs in WA. The analysis was performed for both the wet (JAS) and dry (DJF) seasons and focused on two climatically contrasting areas in the region (i.e., Guinea and Sahel). We have first identified events of LLC occurrence





in both regions. We then analyzed some regional- and local-scale environmental conditions which occur during those events
of LLC occurrence. This analysis mainly focused on horizontal moisture transport by atmospheric circulations, vertical
motions, and surface heat fluxes. Finally, the attenuation of incoming shortwave radiation during the occurrence of LLC in the
region was estimated. To account for the influence of fractional coverage of LLC on the attenuation of incoming shortwave
radiation, we classified the events of LLC occurrence into two groups: events with low fractional coverage (LLC Class-1) and
events with high fractional coverage (LLC Class-2). The main findings of the study are summarized below:

- The occurrence frequency of daytime LLCs is characterized by significant diurnal and seasonal variations: (i) Over
Guinea, the occurrence of LLC Class-1 events is lower during the core of the monsoon season (JJAS) while LLC
Class-2 events are much higher during this period. During the day, LLC Class-2 events occur frequently during the
early morning while LLC Class-1 is rather frequent during the late afternoon. (ii) Over Sahel, the occurrence of both
LLC Class-1 and LLC Class-2 events is higher during the core of the monsoon season. Occurrence frequency of LLC
Class-2 events is higher around midday while for LLC Class-1 events, it does not show a strong diurnal variation.

- During JAS, the occurrence of LLCs (both Class-1 and Class-2) in both Guinea and Sahel is associated with an influx
of cold moist air transported from Guinean Coast by the strong southwesterly winds. This horizontally advected cold
moist air cools and enhances saturation of water vapour and thus leads to the formation of clouds in the region.

- In the dry season, the horizontal advection of moisture does not seem to be related to LLC occurrence (except for a
weak advection in Guinea for LLC Class-2 events) as the region is dominated by dry northeasterly winds during this
period. Instead, turbulent upward motions, likely triggered by the convergence of air masses near the surface appear
to be associated with the occurrence of LLC events. Other processes such as the atmospheric waves and jets in the
region (African Easterly wave, African Easterly Jet, etc.) not specifically analyzed in this study may have also
contributed to the occurrence of LLCs.

- Surface heat fluxes play a major role in the occurrence of LLC Class-2 events than for LLC Class-1 events.
Furthermore, the role of the surface heat fluxes appears to be more important for LLC occurrence in Sahel than in
Guinea. For LLC Class-1 events in Sahel, the surface sensible heat flux appears to be very important for LLC
occurrence during the dry season (sensible heat anomaly of about 25 W m$^{-2}$ in January). This perhaps explains the
turbulent upward air motions that associated with the occurrence of LLCs in DJF.

- Attenuation of incoming shortwave radiation is higher during JAS than DJF and depends on the fractional coverage
of LLCs. In JAS, the occurrence of LLC Class-1 leads to mean attenuations of 29.9 % and 17.5 % respectively in
Guinea and Sahel. LLC Class-2 leads to mean attenuations of 49.1 % and 44.2 % in Guinea and Sahel respectively.
In DJF, mean attenuations of 13.3 % and 32 % are experienced in Guinea for LLC Class-1 and Class-2 events
respectively. In Sahel, the occurrence of LLC Class-1 events leads to a 16.3 % attenuation of incoming shortwave
radiation.



By using the ERA5 cloud cover dataset, previously evaluated in Danso et al., (2019), we provide important information on daytime LLCs which have not been studied in WA – a region notably known for its lack of surface observations. Previous studies have mostly focused on nighttime LLCs in the southern part of WA and only during the WAM season. The study helps

to identify some conditions such as horizontal moisture advection, turbulent upward motions, the convergence of air masses as well as surface heat transfer that occur in association with the occurrence of LLCs. The results on the seasonal and diurnal distributions of LLCs as well as on that on the attenuation of incoming shortwave radiation during LLC occurrence could be useful when making feasibility studies for large-scale solar energy projects.

**Appendix A**

During DJF, positive anomalies of specific humidity (Fig. A1) are related to the formation of LLCs in Sahel (Class-1) and Guinea (Class-2). In both regions, the positive anomalies appear to be as a result of moisture influx from the North Atlantic Ocean aided by northwesterly wind anomalies. These positive moisture anomalies combined with the near-surface processes such as turbulent upward motion due to surface heating and convergence of air masses play contribute significantly to the occurrence of LLCs during the dry season in both regions.


**Author contributions.** DKD, SA, AD fixed the analysis framework. DKD carried out all calculations, produced the figures and wrote the first draft. KK and ATK contributed to the interpretation of the figures. All authors contributed to the analyses and to the redaction.

**Competing interests**. The authors declare that they have no conflict of interest.

**Acknowledgements.** We thank ECMWF for providing ERA5 reanalysis. We are grateful to Institute of Environmental Geosciences (IGE; University of Grenoble – Alpes, France) and to Laboratoire de Physique de l'Atmosphère et de Mécanique des Fluides (LAPA-MF; University Felix Houphouet Boigny, Côte d'Ivoire) who hosted DKD during his stay in Grenoble and

Abidjan in the framework of the International joint laboratory NEXUS on climate, water, land, energy and climate services (LMI NEXUS).

**Financial support.** This research has been supported by the France National Research Institute for Sustainable Development IRD (Institut de Recherche pour le Développement, France).






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

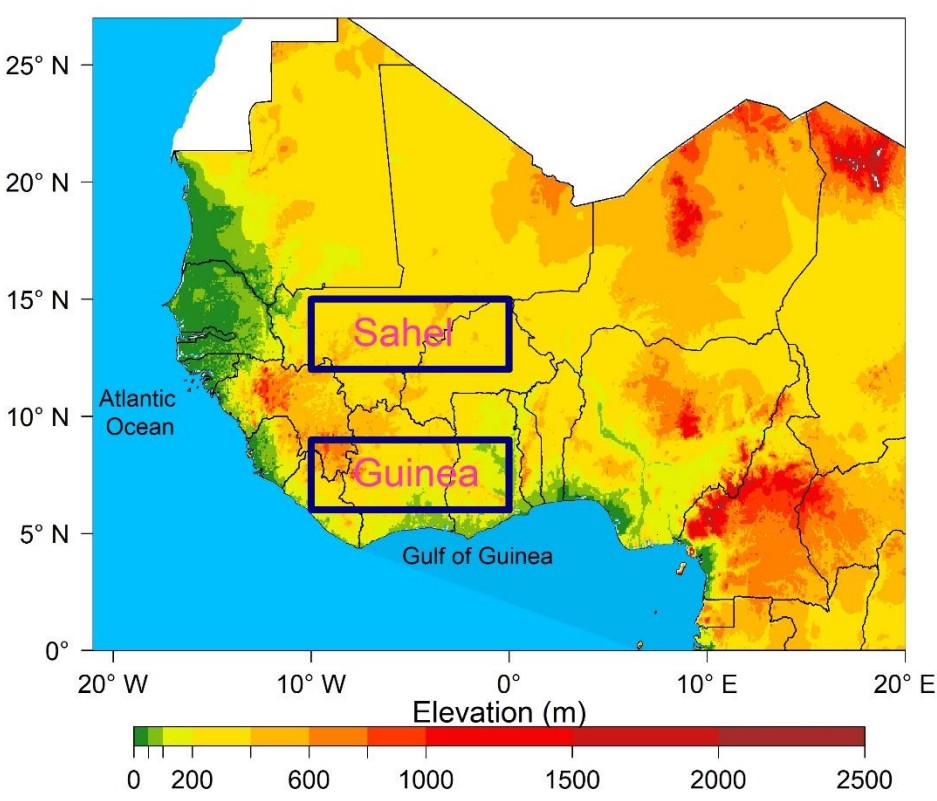

**Figure 1: Map of West Africa showing the two selected windows (dark blue rectangles) used for identification of LLC events. The name denoted for each window is based on the climate zone of the window in the region.**





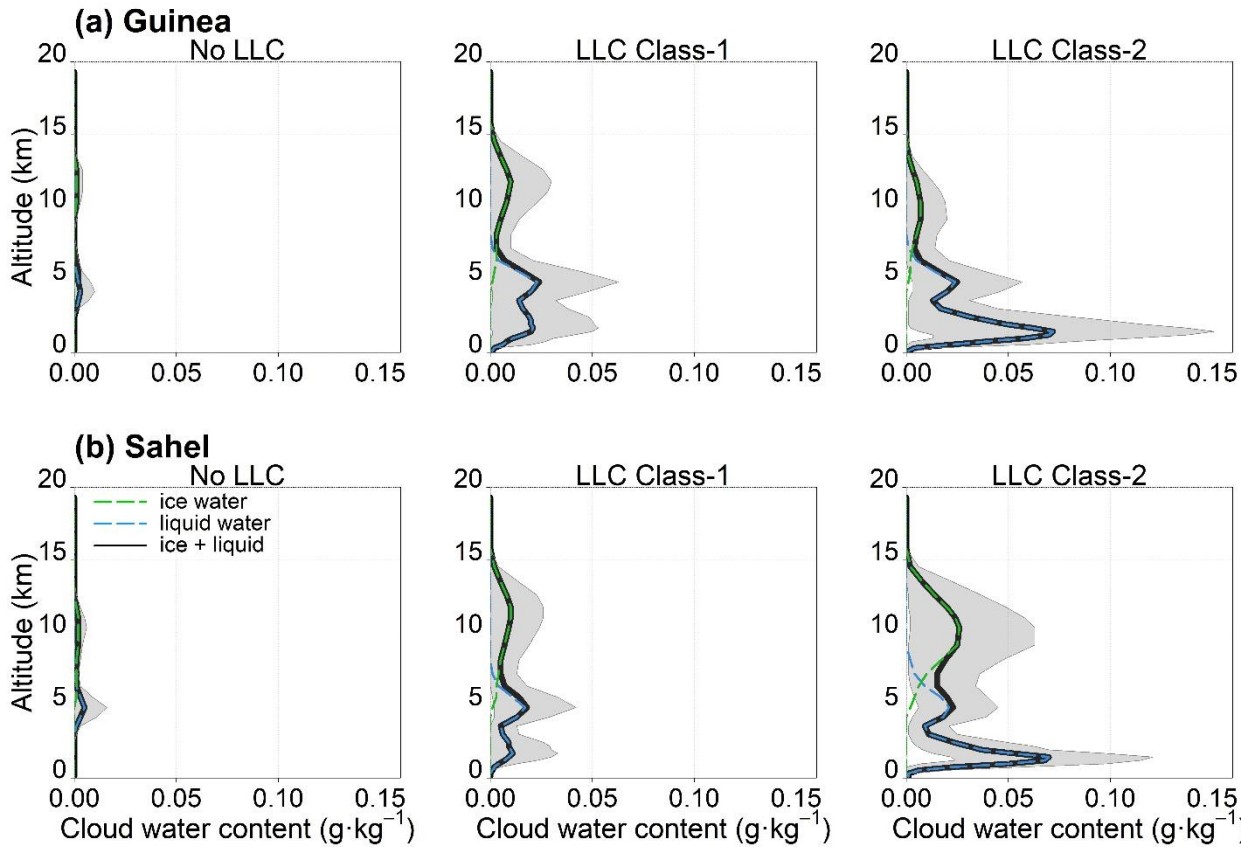

**Figure 2: Mean vertical profiles of cloud water content (ice water (dotted green line), liquid water (dotted blue line), and total water content (black line)) during the occurrence of each LLC class in Guinea (top) and Sahel (below). Shaded areas refer to the 90th and 10th inter-percentile range.**






**Figure 3: Diurnal distribution of occurrence frequency of the three daytime LLC classes during the daytime in the Guinea and Sahel regions as defined in Fig. 1.**





**Figure 4: Monthly distribution of the occurrence frequency of hourly events for the three daytime LLC classes during the daytime in the Guinea (green bar) and Sahel (orange bar) regions defined in Fig. 1. The shorter black bars are the distribution of occurrences that last for at least six consecutive hours (the actual value has been multiplied by a factor or 4 to improve the legibility of the distribution).**


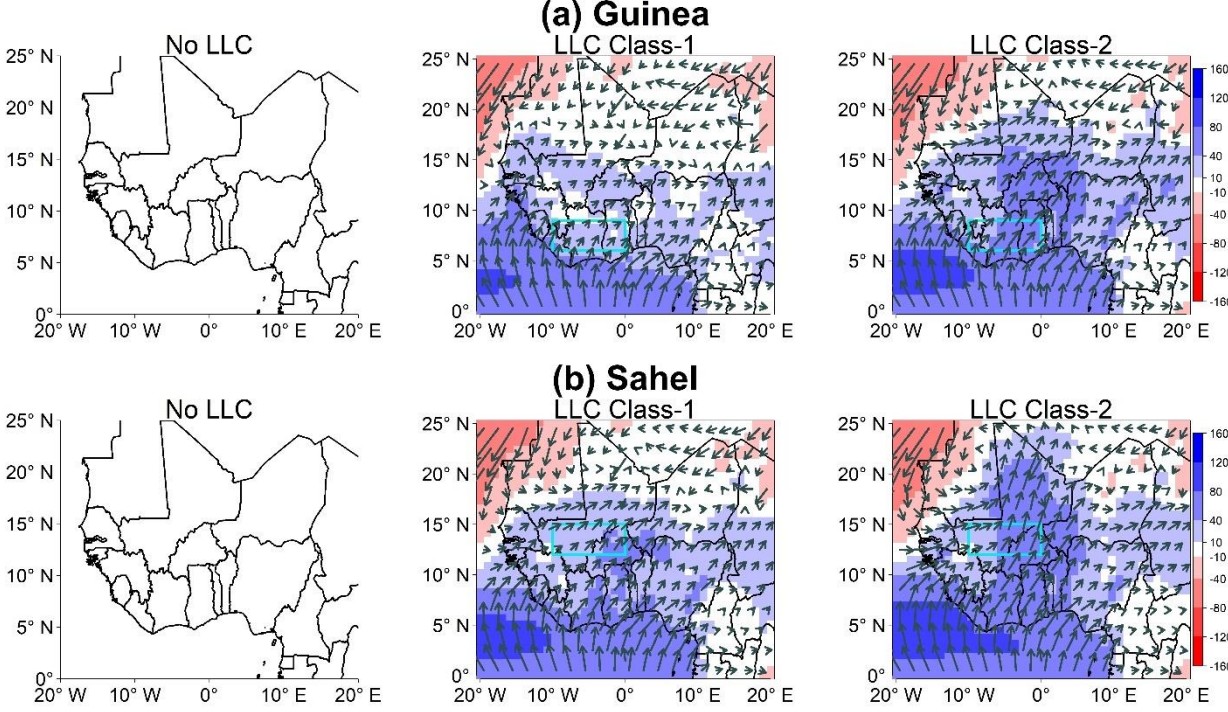

**Figure 5: Composites of 950 hPa horizontal advection of moisture (colours, $Q_{adv}$ in g·kg⁻¹·ms⁻¹) and wind (vectors, m s⁻¹) over WA for the three LLC occurrence classes in the Guinea (top) and Sahel (below) windows during JAS from 2006 to 2015. Only occurrences that last for at least six consecutive hours are considered. Wind vectors are averaged over 2 grid points in both the x and y directions. No event of the No LLC class has been extracted in the Guinea and Sahel regions during JAS, hence the empty plots.**


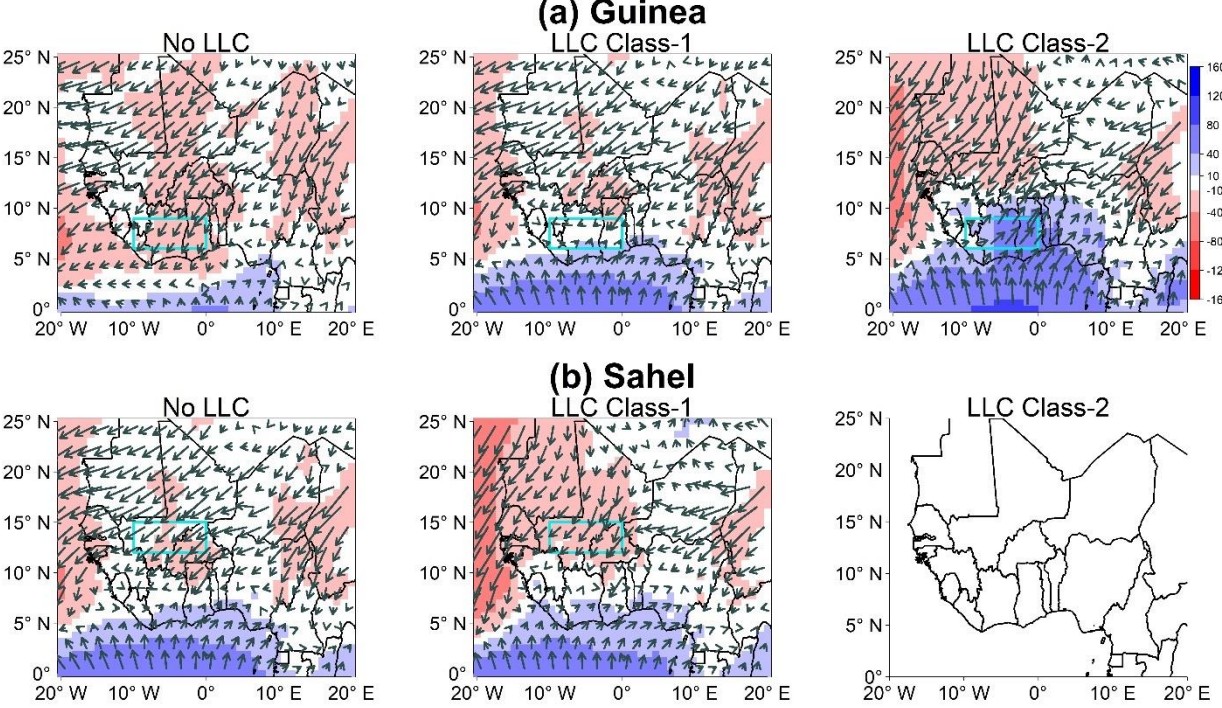

Figure 6: Same as Fig. 5 but for DJF. No event of LLC Class-2 has been extracted in the Sahel area during DJF, hence the empty plot.


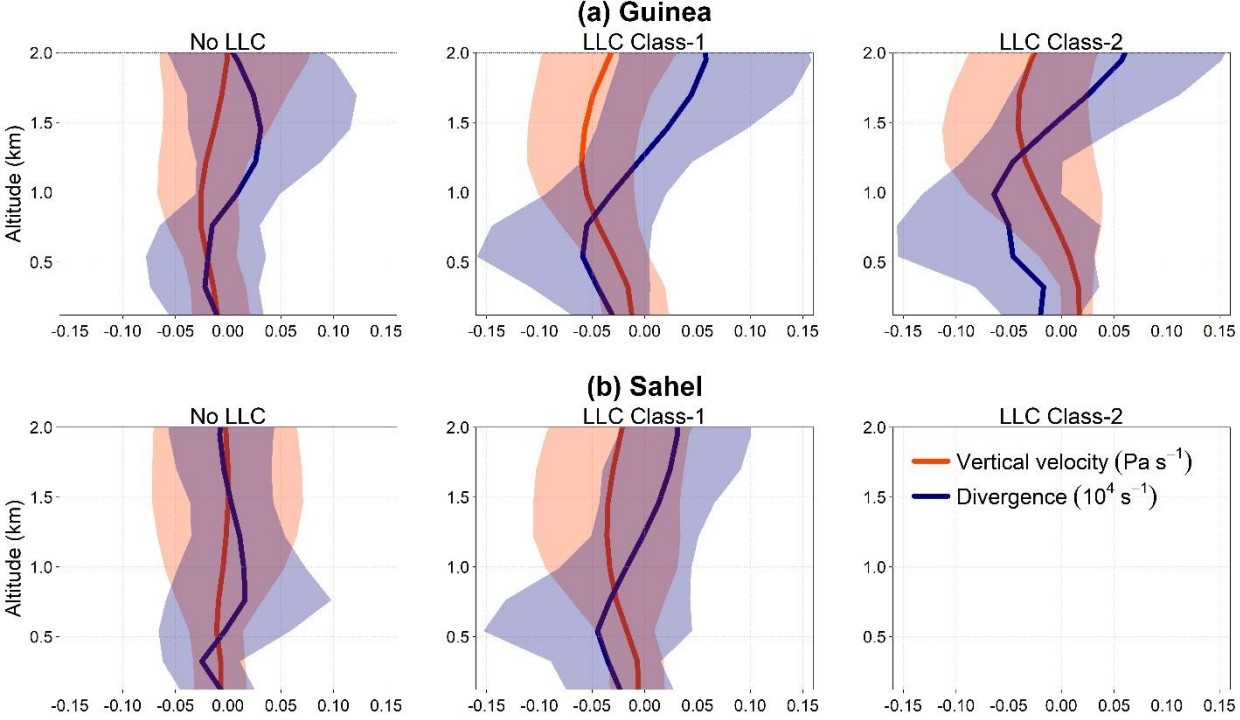

**Figure 7: Composite mean vertical profiles (solid curves) of horizontal air divergence and vertical velocity from the surface to 2 km altitude in the Guinea (top) and Sahel (below) regions for the three LCC occurrence classes during DJF from 2006 to 2015. Divergence has been multiplied by $10^4$ in order to have a similar scale as vertical velocity. Shaded areas refer to the $10^{th}$ and $90^{th}$ inter-percentile range. No event of LLC Class-2 has been extracted in the Sahel window during DJF, hence the empty plot.**





**Figure 8: Monthly mean anomalies of (a) surface sensible heat and (b) surface latent heat fluxes computed in the Guinea and Sahel regions for the occurrence of the three LLC classes from 2006 to 2015. The orange and green stars mean that there was no occurrence of the particular LLC class events in the region during that month, hence the composites were not computed. Only occurrences that last for at least six consecutive hours are considered.**





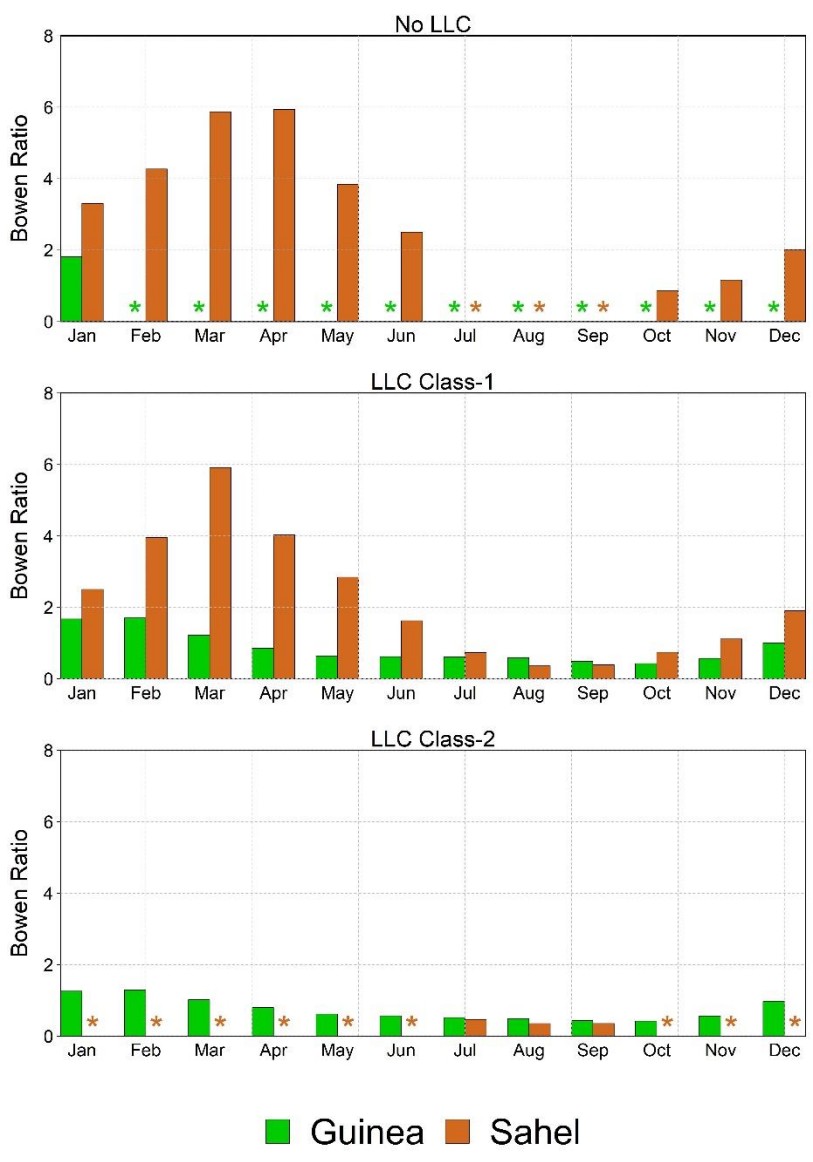

**Figure 9: Monthly mean Bowen Ratio computed in the Guinea and Sahel areas for the occurrence of the three LLC classes from 2006 to 2015. The orange and green stars mean that there were no occurrences of the particular LLC class in the region during that month, hence the composites were not computed. Only occurrences that last for at least six consecutive hours are considered.**



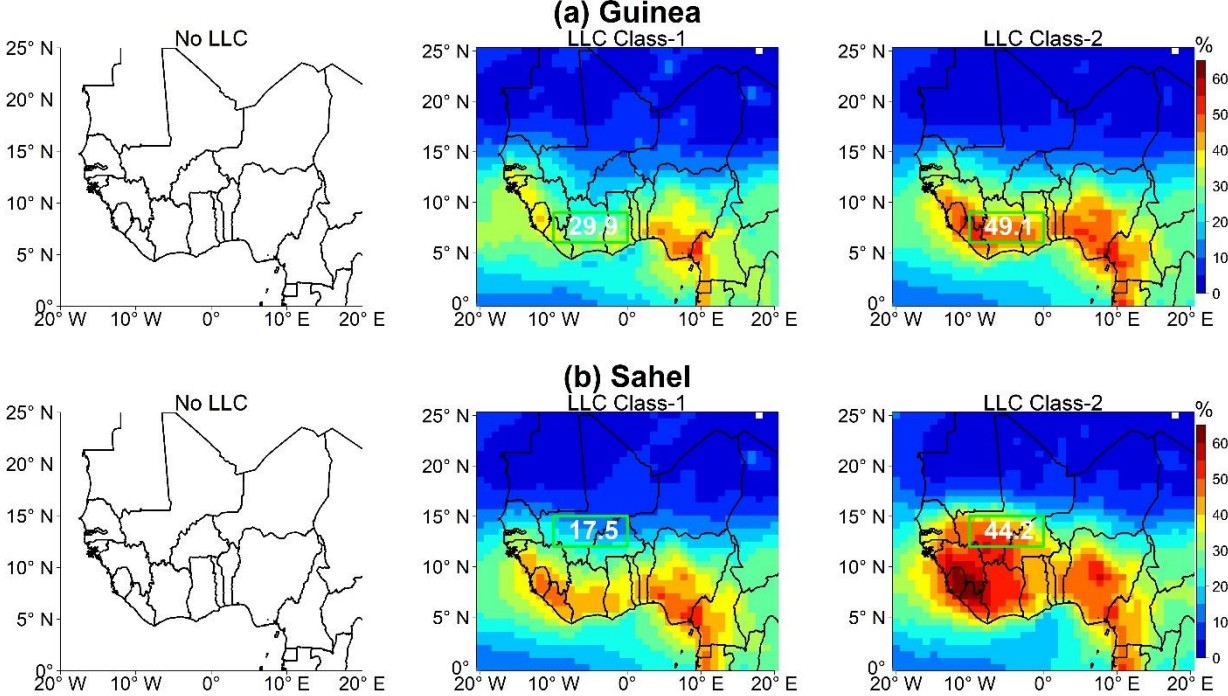

**Figure 10: Mean attenuation of incoming downwelling shortwave radiation (%) during JAS from 2006 to 2015 over WA for occurrence of the three LLC classes in the Guinea (top) and Sahel (below) regions. Only occurrences that last for at least six**
615 **consecutive hours are considered. Numbers in the box refer to the mean percentage attenuation of incoming shortwave radiation in that specific area. No event of the No LLC class has been extracted in the Guinea and Sahel regions during JAS, hence the empty plots.**





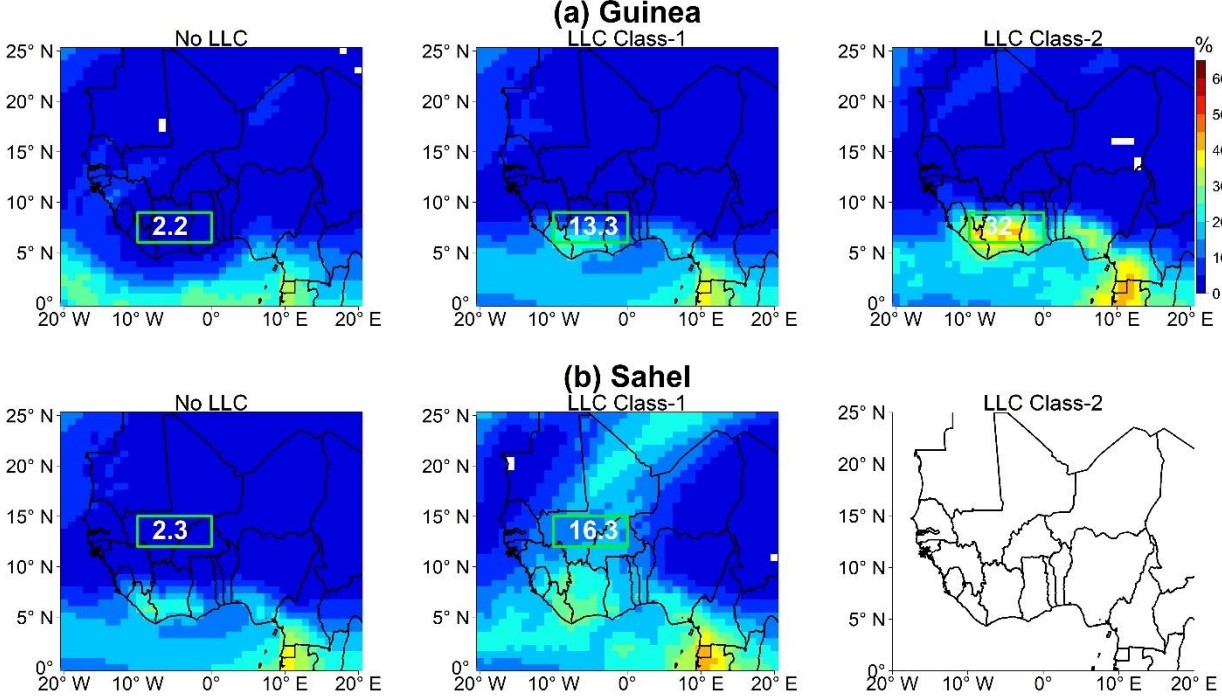

**Figure 11: Same as Fig. 10 but for DJF. No event of LLC Class-2 has been extracted in the Sahel region during DJF, hence the empty plot.**



**Table 1: Definitions of the three classes of LLC occurrence based on cloud fraction.**

| Class of LLC occurrence | Description |
| --- | --- |
| No LLC | $CF = 0$ |
| Class-1 | $0 < CF \leq 0.5$ |
| Class-2 | $CF > 0.5$ |

625



**Table 2: Number of events found for the occurrence of the three LLC classes in JAS and DJF. Composite analysis is performed based on these events for each class and season.**

|  | JAS | | DJF | |
|  | Guinea | Sahel | Guinea | Sahel |
| --- | --- | --- | --- | --- |
| Number of No LLC events | 0 | 0 | 0 | 876 |
| Number of LLC Class-1 events | 178 | 824 | 888 | 12 |
| Number of LLC Class-2 events | 812 | 24 | 178 | 0 |
| Total number of hours in season | 11040 | | 10800 | |

630



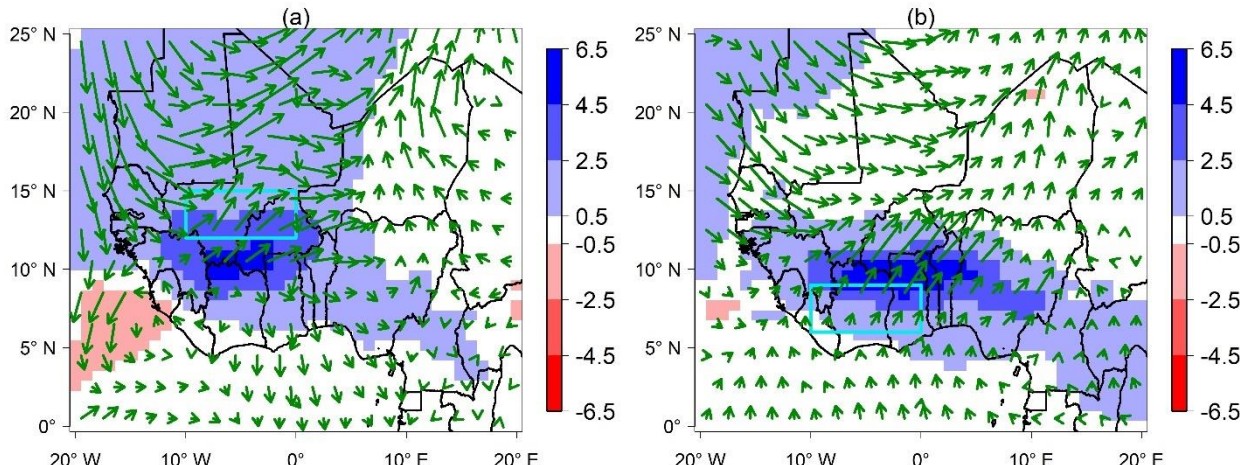

**Figure A1: Composite of anomalies of specific humidity (colours) and wind direction (vectors) for LLC Class-1 in Sahel and LLC Class-2 in Guinea during DJF. Only occurrences that last for at least six consecutive hours are considered. These positive anomalies of moisture in both the Sahel (a) and Guinea (b) regions, seem to help in the formation of clouds during DJF.**