# Peer review of "Daytime low-level clouds in West Africa – occurrence, associated drivers and shortwave radiation attenuation"

_Earth System Dynamics, 2020_

## Referee Comment (RC1) · Anonymous Referee #1 · 22 Jul 2020

REVIEW

**Daytime low-level clouds in West Africa – occurrence, associated drivers and shortwave radiation attenuation**

*Derrick K. Danso, Sandrine Anquetin, Arona Diedhiou, Kouakou Kouadio, Arsène T. Kobéa*

This study, based on the analysis of the ERA5 hourly data from 2006 to 2015, shows the occurrence of daytime low level clouds in WA at two different latitudes (Sahelian and Guinean). It also aims at determining the atmospheric conditions for different LLC classes (based on cloud fraction) and their impact on the solar incoming radiation. This article can be improved at several points of view: motivation, methodology explanation, results interpretation.

Main comments:
(1) The LLC events are selected in this study with their cloud base height (< 2 km). This is in accordance with the definition of low cloud. However, the authors apply this definition at two very different places in Africa and at very contrasting seasons. The boundary layer varies from few hundred meters (during monsoon season in the Guinean region) to several kilometres (during winter season in the Sahelian region). Consequently, the LLC as defined by the authors may include or not boundary layer clouds according to the region and the season. This should be at least discussed (see specific comments) since it impacts consequently the different statistics presented in this study.
(2) In addition to the previous comment, LLC with base below 2 km gather stratus, stratocumulus, cumulus and MCS (as mentioned in the paper). These clouds develop in quite different atmospheric conditions (specifically in terms of divergence). What is the interest of highlighting the atmospheric conditions of a mix of different clouds (see specific comments)?
(3) The cloud fraction as a parameter to define different LLC classes in order to check their impact on solar radiation seems quite logical, but not to determine the atmospheric conditions of these classes.
(4) Some explanations are missing to fully understand the analyzed atmospheric conditions, specially the surface flux anomaly.

Specific comments:
(1) P2, line 40: Perhaps the authors could add a reference to the work of Shrage et al. who first quantified the low level cloud frequency and fraction in southern WA.
  • Schrage, J. M. and Fink, A. H.: Nocturnal continental low-level stratus over tropical West Africa: observations and possible mechanisms controlling its onset, Mon. Weather Rev., 140, 1794–1809,2012.

(2) P2, line 48-57: Three very recent articles published in ACP (DACCIWA special issue) should be cited here since they focus on daytime phase of low level clouds. The first one estimates, with local measurements, the impact of LLC on the surface net radiation and on the convective surface fluxes which is one objective of the present study.
  • Lohou, F., Kalthoff, N., Adler, B., Babic, K., Dione, C., Lothon, M., Pedruzo-Bagazgoitia, X., and Zouzoua, M., 2020 : Conceptual model of diurnal cycle of low-level stratiform clouds over southern West Africa. Atmos. Chem. Phys., 20, 2263–2275, doi: 10.5194/acp-20-2263-2020.
  • Maurin Zouzoua, Fabienne Lohou, Paul Assamoi, Marie Lothon, Véronique Yoboue, Cheikh Dione, Norbert Kalthoff, Bianca Adler, Karmen Babić, and

Xabier Pedruzo-Bagazgoitia : Breakup of nocturnal low-level stratiform clouds during southern West African Monsoon Season, https://doi.org/10.5194/acp-2020-602

- Pedruzo-Bagazgoitia, X., de Roode, S. R., Adler, B., Babić, K., Dione, C., Kalthoff, N., Lohou, F., Lothon, M., and Vilà-Guerau de Arellano, J., 2020: The diurnal stratocumulus-to-cumulus transition over land, Atmos. Chem. Phys., https://doi.org/10.5194/acp-2019-659.

(3) P2-P3 : I do not understand the link between the last sentence of page 2 and the first one of page 3. Could the authors be more explicit? Or perhaps the word « consequently » is not appropriate in this sentence?

(4) P3, Line 3 : « ...have been limited to the WAM season... »

(5) P3, line 70 : « Few studies which were done with simulations, reanalysis, and satellite data in the region (e.g., Adler et al., 2017; Knippertz et al., 2011; van de Linden et al., 2015; Schuster et al., 2013) have nevertheless shown results similar to those of ground observational studies to some extent. » Could the authors precise to what observational studies they referred to?

(6) P4, line 100-102: Could the author reword this sentence? Remove this part perhaps "and their surface heat fluxes explore » ?

(7) P5, L127-134: This paragraph about upper clouds is very important at several points of view. Because the threshold on the base height (< 2 km) can mix a lot of cloud types (Stratus, Stratocumulus, Cumulus, MCS) I wonder if it would be interesting to add a statistic on the LLC top heights just to be aware of the cloud types mixed in the so large family of LLC.

(8) P5, equ. 1:
- It seems to me that equation 1 gives the Moisture Flux. The moisture advection would be defined with the horizontal gradient of specific humidity. Qavd is named differently along the article: horizontal moisture flux advection, moisture advection,…
- The sum should be between 1 and N and not between i and N.
- Why the chosen level for the moisture flux estimate is 950 hPa? Could the authors justify this choice? Why the integrated value over the vertical is not used in this study?

(9) P5, L: Ek and Holstag study focuses on ABL clouds.
- The present study does not take into account all ABL clouds since a large part of them are higher than 2 km when the ABL is higher, which is very often the case in Sahelian region.
- Some Stratus or Stratocumulus low cloud are decoupled from the surface and are then not influenced by the surface flux (see previous suggested papers of Zouzoua et al., Pedruzo et al, and Lohou et al.).
➔ These two points should be addressed in some way.

(10)      P6, Equation 2 & 3:
- CRE has two different definitions through equation 2 & 3. The authors could remove equation 2 and just keep equation 3.

- What are the clear sky events? Are they cases with no liquid water at all in ERA5? If so, how many cases are used for CRE estimate? Or are they theoretically calculated? This is important to understand the CRE for no-LLC cases.

(11)  P6, L163: "...$SW_\downarrow$ is the downwelling shortwave radiation in all-sky conditions..." If I understand correctly $SW_\downarrow$ is rather the downwelling short wave radiation for each LLC classes instead of "all sky conditions". This could be written in first place?

(12)  P6, L169: The authors should make clear in the text and the legend that this daytime distribution gathers all the seasons.

(13)  P6, L180: "Additionally, the early morning peak in the events of LLC Class-2 could also be partly linked to contributions from tropical oceanic low-level convection which is maximum during the early morning (Yang and Slingo, 2001)." Could the authors check the Yang and Slingo paper? It seems to me that Yang and Slingo mentioned the "Tropical oceanic deep convection" (for example page 798-799 in Yang and Slingo, 2001). If this is so, the paragraph the authors certainly referred to finishes this way : "Some of these convective systems, under optimal environmental conditions, continue to grow and reach their mature stage some time later during the night and early morning". It is rather deep convection and MCS that Yang and Slingo are talking about. How do we know that LLC class 2 are deep convection if all seasons are mixed in fig 3? What is the proportion of deep-convection versus stratocumulus? In what extent the MCS can impact the statistic presented in Fig 3 for Guinean region? Did the authors try a diurnal cycle for each season?

(14)  P7, All comments on Figure 3: I don't know what the authors call "well-marked diurnal cycle" or "weak diurnal cycle"? For example, 10% of 2500 cases is a larger variation (LLC class-1 Sahelian region) than 4% of 200 cases (LLC class-2 sahelian region).

(15)  P7, L189-192: I agree with this comment on the type of cloud included in the statistic. That helps to understand what really means this composite diurnal cycle.

(16)  P7, L193-214:
- I think these two paragraphs should be moved at the beginning of this section. It would be interesting for the reader to have a first insight of the seasonal variation before the diurnal one.
- The threshold of 2 km for the cloud base might impact the Sahelian region distributions. The number of no LLC class would decrease with a higher cloud base height threshold whereas LLC class-2 number would increase. That means that MCS, for example, can be sorted out as no-LLC class when ABL is higher than 2 km. Is that possible? In what extent this 2-km threshold impacts the atmospheric conditions for the different classes?

(17)  P8, L228, Table 2:
- The total number of hours in season indicated in Table 2 should be presented and commented in the legend and the text?
- There is zero No-LLC events detected in Guinean region during DJF season according to table 2. But there are atmospheric conditions for this class in figures 6 and 11 for example. I guess the table is wrong.

(18)  P8, L231: "...cold moist air from the ocean associated with the strong southwesterly winds (positive $Q_{ad}$)..." Figure 5 and 6 are very nice. I am not used in analysing the moisture flux but

according to the Figures 5 and 6, $Q_{adv}$ seems rather dominated by the wind intensity than the moisture (and even less the temperature of course) from my point of view. I think, and some AMMA and DACCIWA studies show this, that the moisture advection is null in the Guinean region during the WAM (e.g. Adler et al., Babic et al.), whereas it is important in the Sahelian region. It depends on the moisture gradient. So what does the moisture flux show? The monsoon or the Harmattan horizontal expansion according to the season?

(19)     P8, L232: "Predictably, the horizontal advection of moist air is stronger during the occurrence of LLC Class-2 events than LLC Class-1 events." Moisture flux?

(20)     P8, L234," This inland advection of moist air from the ocean has been found to play a major role in cooling (Adler et al., 2019b) which in turn enhances saturation of water vapour and consequently LLC formation (Adler et al., 2019, 2017; Babić et al., 2019b)." Moist air advection cannot induce a cooling. Adler et al., Babic et al. and Lohou et al. showed that there is no moist air advection but cold air advection. It is the cold air advection by the low level jet and Atlantic inflow which induces the saturation and the cloud formation in the Guinean region. That means that the monsoon flow is not moister than the air in the Guinean region where rain events are frequent. This can be different for the Sahelian Region which is much drier than the Guinean one. Please consider to change this comment and make different comments for the two regions.

(21)     P8-9, L245-259: this paragraph deals with the vertical velocity and the divergence of the horizontal wind.
- I would show only one of these parameters since they should be proportional. No need to comment both.
- The authors precise at the beginning that a negative value of these two parameters is favourable to cloud formation. Stratus and stratocumulus are characterized by positive vertical velocity (hPa s$^{-1}$). So this comment cannot be general and this shows the limit of searching for atmospheric conditions of a class which may mix different clouds.
- "It is also important to note that during JAS, the average vertical profiles of these processes (divergence and vertical velocity) are not similar to DJF (not shown).". If this is important, perhaps the authors should comment a little more on this or show the figures for JAS.
- I am not sure what the reader can conclude from this analysis.

(22)     P9-10, section 4.2:
- Could the author precise how the monthly anomaly is computed because I am very surprised by these results?
- The reduction of the net radiation at surface by the clouds induces a reduction of the surface sensible and latent heat flux. So, from my point of view, the figure 8 should rather show the cloud impact on the surface energy budget than the effect of the convective flux on the clouds.
- This is why I do not understand how the author can find a negative anomaly of the sensible heat flux during no-LLC event.
- The anomalies should be discussed in comparison to the flux itself. Do the author think that an anomaly lower than 10 W m$^{-2}$ is significant when the surface flux is around 300 W m$^{-2}$ and considering the error of the surface flux in the model.
- At last, the authors deduce from the surface flux anomaly a surface temperature anomaly. The surface heat fluxes are proportional to the temperature vertical gradient and not directly linked to the surface temperature. Such a discussion is misleading.

(23)     P11, L313-316: I fully agree with this statement. The surface convective flux should be also reduced according to the cloud fraction. So how the authors can detangle the effect of the cloud on the surface flux from the effect of the flux on the cloud triggering?

---

## Referee Comment (RC2) · Anonymous Referee #2 · 28 Jul 2020

Review of

**Daytime low-level clouds in West Africa – occurrence, associated drivers and shortwave radiation attenuation**

by Derrick K. Danso, Sandrine Anquetin, Arona Diedhiou, Kouakou Kouadio, Arsène T. Kobéa

This paper presents the analysis of the daytime low level clouds (LLC) over West Africa using ERA5 dataset for the period 2006-2015 to investigate their occurrence frequency, seasonal and diurnal cycles, as well as the associated atmospheric conditions for the two contrasting regions, i.e. Sahelian and Guinean region. Based on the cloud fraction, three classes were formed (no LLC, LLC class-1, and LLC class-2) and the results indicate that during the summer months LLC class-1 has the peak occurrence frequency in the Sahel, while LLC class-2 is dominant in the Gulf of Guinean. Finally, this study also addressed the attenuation of the shortwave downwelling radiation due to the LLC and found that during the summer months it is on average between 44 % and 49 % for the Sahel and Guinean region.

Considering the importance of the LLC for the regional climate of West Africa and the general lack of studies addressing these clouds, I find the topic of this study to be relevant and suitable for publication in ESD. This study provides new results which in my opinion will be useful, especially considering the ongoing and future solar energy project. However, before this manuscript can be accepted for the publication, it should be carefully revised, since there are some shortcomings in the analysis and the presentation of the results needs to be more clear. My comments and suggestions are listed below.

Major comments:

1.  I do not understand why authors extract reanalysis data of a higher resolution of 0.25o to a very coarse 1o resolution. What do you mean by "directly extracted" (Pg. 3, line 87)? Although the focus is on the analysis of large scale conditions, it is well known that some mesoscale processes are responsible for the formation and maintenance of LLC. Have you performed some sort of validation test to make sure that the conditions are properly presented with the data of coarser resolution? Additionally, sensible and latent heat flux analyzed here are not large scale phenomena.

2.  In my view, the analysis of the atmospheric conditions should be presented more consistently and more clearly.

    The definition of the "horizontal moisture flux advection" is not correct (Pg. 5, line 141). Namely, eq. (1) presents the average moisture flux and not the advection of the moisture! Therefore, the presentation and discussion of the results regarding the advection of the moisture need to be carefully revised. If Figs. 5 and 6. show the quantity calculated according to the eq. (1), then they do not show moisture advection. The large majority of the paper discusses the role of moisture advection, however, this quantity is not calculated.

On Pg. 6 Equation (2) is inconsistent with the text on lines 160-161: shouldn't the equation read as: CREswt=SWt-SWcs(t)?

3. Due to the definition of the LLC used in this study, i.e. all clouds with cloud base below 2 km, different cloud types are considered as LLC. This causes some confusion in understanding atmospheric forcing related to different LLC classes in the two regions. For example, the dominant LLC class-2 during the JAS period in the Guinean region is most likely related to the stratiform clouds and shallow cumulus, while the LLC class-2 peak in the same season most likely corresponds to deep convective clouds (as it corresponds to rain events). These different cloud types have different forcings and the authors need to make a more clear distinction between these throughout the manuscript. This is especially confusing in the Abstract.

   The authors should refer to the recent finding from the DACCIWA regarding the physical processes responsible for the formation and maintenance of the LLC over the Gulf of Guinea during the WAM season.

4. In the recent DACCIWA papers (ACP special issue available at https://www.atmos-chem-phys.net/special_issue914.html), the advection of the cold maritime air mass, related to the low level jet and the SW monsoon flow, is found to be the key process responsible for the LLC formation and not the advection of moist air. On the other hand, the advection of the relatively moist air could be important for the LLC formation in the Sahel. However, the authors do not assess the role of temperature advection. What is the reason for this?

Minor comments:

- Pg. 2, line 59-60: This statement is not correct. In the Guinean region the LLC form during the night and persist long into the morning and early afternoon hours, therefore have a direct impact on the surface solar irradiance. Please see the study of Lohou et al. (2020) and Zouzoua et al. (2020) regarding this.
- Pg. 7, lines 198-200: Here it would be better to refer to recent DACCIWA publications which show that the advection of cold air, and not the advection of moist air, is the key process responsible for the formation of LLC in the Guinean region during the monsoon season.
- Pg. 8, line 234-236: The reference here should be Adler et al. (2019) not (2019b). Additionally, all the studies referenced here find that the advection of cold air is the major factor in leading to saturation, not moist air! The authors should carefully revise the paper when discussing the processes leading to the formation of LLC in the two regions and when referring to previous studies to avoid making mistakes like these. Namely, based on the DACCIWA observational data it became clear that the advection of cold maritime air is the dominant process for the LLC formation in the Gulf of Guinea, while the advection of moist air into the Sahelian region is the most dominant process for LLC formation.
- Pg. 8, line 242: It should be Saharan Heat Low.

- Pg . 8, line 243: What is the role of the cold air advection?
- Pg. 8, line 245: Consider here replacing "moist air" with cold air.
- Pg. 9, line 287: How can water vapor be cooled by moist air advection?
- Pg. 10, line 304 and 312: Which region are you referring to? Is it entire West Africa?
- Pg. 10, line 310: Figs. 10 and 11 also show the downwelling shortwave radiation attenuation for no LLC class.
- Pg. 11, line 329-330: The sentence "In Sahel, the mean attenuation during the occurrence of LLC Class-1 events is 16.3% *though* areas around the southern coast of WA can experience higher losses." should be rephrased since it is not clear what is the connection between the attenuation in the Sahel and the southern coast of WA.

---

## Referee Comment (RC3) · Anonymous Referee #3 · 29 Jul 2020

**Review of the manuscript titled**
**"Daytime low-level clouds in West Africa – occurrence, associated drivers and shortwave radiation attenuation" by Danso et al.**

This manuscript investigates daytime (06-17h) low-level clouds (LLC) over West Africa based on ERA5 (2006–2015) in two regions, the Sahel and Guinea Coastal region. LLC taken from the ERA5 archive describes cloudiness below 800 hPa (~2km). Foci of the study are on diurnal cycles in the dry and wet seasons, the seasonal variation of LLC, the atmospheric conditions related to different classes of LLC and the impact of the latter on the incoming solar radiation.

While the article describes an overall interesting topic and contains some interesting material, it also has weaknesses that shall be addressed in a major revision.

1) There needs to be a proper explanation between ERA-5 cloud fraction that usually depends on the subgrid-scale cloud scheme and the liquid and ice water paths that are relevant for the radiation scheme. Cloud fraction can be larger than zero for relative humidities below 100% and zero liquid or ice water content. There is an interesting discussion on this in Hannak et al. (2017, JCLIM). Thus cloud occurrence frequency at a gridpoint can be defined by a sub-grid scale cloud fraction or a hydrometeor content > 0. There needs to be explanations/discussion of this issue (see below).

2) My major concern that also needs some time in the revision relates to the degree of realism that hourly ERA5 data have in the representation of LLC. There was no evaluation of ERA5 in Danso et al. (2019), only a comparison of total cloud cover (all clouds, all levels) based on a very coarse cloud fraction partitioning of METAR reports for three stations in West Africa. From this one figure in the Suppl. Mat. it is not obvious that the CERES data set would be inferior to ERA5 – which it likely is according to the findings in Danso et al. I am very worried about that this study may show physically consistent errors of the underlying ERA5 model. For example high LLC fractions/frequencies may be related to errors in the Bown ratio in the ERA5 model etc (with causality being another point of concern). The Reviewer proposes two ways out of it: One is to use available ground and satellite observational evidence that exists, the other is to use two other re-analyses (e.g. MERRA2, JRA-55, NCEP). In terms of the former, van der Linden et al. (2015) have shown the usefulness of the 2B-GEOPROF-LIDAR tracks for cloud occurrence frequency in mean, 250m vertically resolved profiles, including the layer below 2 km (their Figure 6). The sampling argument given in the manuscript is not robust, as is the argument of the 1x1 grid resolution needed for PV application – the purpose is to validate the usefulness of ERA5 cloudiness and for this this the combined Cloudsat-Calipso at 01:30 LT would serve the purpose. Moreover, multi-year measurements of solar incoming radiation are available from AMMA-CATCH (http://www.amma-catch.org/?lang=fr), for the Upper Ouémé site even multiyear measurements of sensible and latent heat fluxes can be obtained. Radiation measurements for Parakou and Cotonou are also available from doi: 10.6096/baobab-dacciwa.1785. Kniffka et al. (2019, ACP) have shown large errors in surface solar

radiation in ERA-I and it is questionable that this has been improved a lot in ERA5. Kniffka et al. (2020, QJRMS) have also shown that short-term forecasts of weather forecasting models, among which is the ECWMF IFS model have large errors in precipitation, radiation, and cloud cover. So there are strong arguments to validate ERA5 before drawing far reaching conclusions. I prefer to use the few, but available observational evidence, but using two other reanalyses also allows inferences about the fidelity of the results. Clearly, I do not want the author to go to deep into validation, but some more validation is necessary (for some observational points and subperiods of 2006–2015).

3) LLCs and MCSs
Having been in the region many times, I can't understand why MCSs should explain a large fraction of LLCs in the Sahel, for example. There are LLCs in the "small" leading edge/in the convective part of the MCSs, but not in the stratiform part. And MCSs are relatively infrequent. LLCs in the rainy season over the Sahel occur in the morning, but dissipate in the afternoon when isolated Cu cong or CB develop. Please comment on this.

4) I have not corrected all language errors and some statements are not very clear. The author should go over the manuscript meticulously in the revision to account for this deficiency.

Minor comments:

l. 29: "efficiently represent convection AND CLOUDINESS"

l. 35: "escalation" is not the right wording

l. 37: "remains"

l. 40: prefer references in a chronological order.

l. 40: "van deR Linden", please correct.

l. 40: "Farther north"

l. 43: "persisting into the early afternoon"

l. 43: "as shallow convective clouds"?????

l. 45-47: Here a reference to Kniffka et al. (2019, ACP) is appropriate

l 50: "The majority if these studies"

l. 63: "limited TO the WAM season"

l. 84: "A better reference to ERA5 is now Hersbach et al. (2020, doi: 10.1002/qj.3803)

l. 101 "their surface heat fluxes explore", awkward sentence

l. 147 "horizontal air divergence", omit "air", perhaps add "wind".

l. 182 Reword "convected air"

l. 188-192: Doubt that MCS contribute to LLC in reality (see major comment 3)

l. 202-204: What about the contribution of morning LLC?

l. 212: Did Mathon et al. (2002b) really refer to LLC below 2km?

l. 232: the adverb "predictably" seems inappropriate here. Please rephrase.

l. 234 "...cold moist". Sentence terminates awkward.

l. 240-242: q anomalies transported from the Atlantic in DJF and modulation by the WA heat low? The DJF heat low is somewhere over the Central African Republic/South Sudan at this time of the year. Please clarify.

l. 248: Very awkward explanation of divergence. Usually, the divergence of the 2-d wind field is a good approximation of mass divergence since horizontal gradients of density are small. Please rephrase.

Section 5: The liquid and ice water content is relevant for attenuation. I am pretty sure it is not the subgrid scale cloud fraction (please clarify this point here or in the data section, see major comment 1).

l. 363-365: "Other..." these processes or better features are only relevant in the wet season, not the dry season. Please mention this here.

l. 393 "redaction" I think this is not the right word.

---

## Author Comment (AC1) · 24 Aug 2020

**Responses to Referee 1**

Daytime low-level clouds in West Africa – occurrence, associated drivers and shortwave radiation attenuation *by Danso et al.*

On behalf of all the authors, I wish to thank the reviewer for the thorough assessment of our study and for providing us with comments to improve the manuscript during the revision. Please see below the detailed responses to each of the reviewer's comments/questions. The reviewer's comments are shown in black font while our responses are shown in blue font. Where applicable, the changes that will be made in the manuscript are shown in italics.

This study, based on the analysis of the ERA5 hourly data from 2006 to 2015, shows the occurrence of daytime low level clouds in WA at two different latitudes (Sahelian and Guinean). It also aims at determining the atmospheric conditions for different LLC classes (based on cloud fraction) and their impact on the solar incoming radiation. This article can be improved at several points of view: motivation, methodology explanation, results interpretation.

We thank the reviewer for the positive and constructive comments to improve our work.

Main comments:

(1)    The LLC events are selected in this study with their cloud base height (< 2 km). This is in accordance with the definition of low cloud. However, the authors apply this definition at two very different places in Africa and at very contrasting seasons. The boundary layer varies from few hundred meters (during monsoon season in the Guinean region) to several kilometres (during winter season in the Sahelian region). Consequently, the LLC as defined by the authors may include or not boundary layer clouds according to the region and the season. This should be at least discussed (see specific comments) since it impacts consequently the different statistics presented in this study.

We thank the reviewer for this comment. We fully agree that the clouds analyzed in our work may or may not include all boundary layer clouds due to the definition of LLCs. In the revised manuscript we will discuss this important issue which could influence the interpretation of some of our results. The following sentences, for instance, will be incorporated in the methodology section:

*Due to the definition of LLCs used, the present study does not consider the impact of the atmospheric boundary layer (ABL) that may considerably modify the altitude of the low-level cloud. Indeed, ABL clouds have their bases higher than 2km especially in the Sahel region during the dry season. Again, some studies (e.g., Lohou et al. (2020), Pedruzo et al. (2020), Zouzoua et al. (2020)) have shown that some LLCs can be decoupled from the surface and are therefore not influenced by the surface heat fluxes.*

(2)    In addition to the previous comment, LLC with base below 2 km gather stratus, stratocumulus, cumulus and MCS (as mentioned in the paper). These clouds develop in quite different atmospheric conditions (specifically in terms of divergence). What is the interest of highlighting the atmospheric conditions of a mix of different clouds (see specific comments)?

We agree with the reviewer that different low-level clouds may develop under different atmospheric conditions. This question somehow mirrors comment #4 by Reviewer 2. Due to the nature of the ERA5 data, it is impossible to make the distinction between the different types of low-level clouds. However, while low clouds can be associated with different types of forcing (for example in terms of divergence or cold air advection for stratiform clouds as found in the DACCIWA studies), we assume that the moisture flux (which is analyzed) conditions will not differ significantly from one LLC type to the other. In terms of divergence, we believe we can use our results to postulate the clouds present. This is explained below:

In the original manuscript, we showed divergence and vertical velocity for DJF (but not JAS) which showed mostly negative values favorable for LLC formation. In specific comment #21 in this document, the reviewer mentioned that St and Sc clouds are characterized by positive vertical velocity (contrary to our results for DJF) – and therefore this result cannot be generalized for all low clouds, we fully agree with this point. LLCs during DJF are likely not St and Sc clouds but perhaps cumulus clouds. The divergence (and vertical velocity) during JAS is shown in Figure 1 below. As shown, the mean profiles are almost neutral. This can be explained by the fact that in JAS, St, Sc, Cu, and Cb clouds are all present – since they have opposing vertical velocities, their combined effect leads to a mean value almost neutral, as shown.

In the revised manuscript (specifically in discussion section under the discussion of atmospheric conditions), we will introduce and discuss this figure (for JAS) suggesting the types of LLCs that maybe dominant in which season, so that the divergence/vertical velocities will not be generalized as we did previously.

[Figure]

*Figure 1: Vertical profiles of divergence and vertical velocity during occurrence of the LLC classes in JAS*

(3)  The cloud fraction as a parameter to define different LLC classes in order to check their impact on solar radiation seems quite logical, but not to determine the atmospheric conditions of these classes.

We thank the reviewer for this comment although we believe these classes could be also be important for some of the atmospheric conditions. For example, we showed that the Class 2 LLC is associated with a much stronger moisture flux compared to the Class 1. This is however not the case for divergence. In the revised manuscript, we will produce the divergence plots for No LLC

events and all LLC events (so we will not show Class-1 or Class-2). As an example, we show one of the plots in Figure 2:

[Figure]

*Figure 2: Composite vertical profile of divergence for No LLC and LLC occurrences in (a) Guinea and (b) Sahel for DJF.*

(4)  Some explanations are missing to fully understand the analyzed atmospheric conditions, specially the surface flux anomaly.

This has been noted. We are grateful to the reviewer for this comment. We will update the manuscript with further explanations to understand the analyzed surface flux anomalies specifically how the anomalies were calculated and analyzed. Other questions pertaining to the surface fluxes are addressed in our responses to the specific comments.

Specific comments:

(1)  P2, line 40: Perhaps the authors could add a reference to the work of Shrage et al. who first quantified the low level cloud frequency and fraction in southern WA.

  • Schrage, J. M. and Fink, A. H.: Nocturnal continental low-level stratus over tropical West Africa: observations and possible mechanisms controlling its onset, Mon. Weather Rev., 140, 1794–1809,2012.

Schrage et al. (2012) will be referenced in this sentence in the revised manuscript.

(2)  P2, line 48-57: Three very recent articles published in ACP (DACCIWA special issue) should be cited here since they focus on daytime phase of low level clouds. The first one estimates, with local measurements, the impact of LLC on the surface net radiation and on the convective surface fluxes which is one objective of the present study.

  • Lohou, F., Kalthoff, N., Adler, B., Babic, K., Dione, C., Lothon, M., PedruzoBagazgoitia, X., and Zouzoua, M., 2020 : Conceptual model of diurnal cycle of low-level stratiform clouds over southern West Africa. Atmos. Chem. Phys., 20, 2263–2275, doi: 10.5194/acp-20-2263-2020.

  • Maurin Zouzoua, Fabienne Lohou, Paul Assamoi, Marie Lothon, Véronique Yoboue, Cheikh Dione, Norbert Kalthoff, Bianca Adler, Karmen Babić, and Xabier Pedruzo-Bagazgoitia :

Breakup of nocturnal low-level stratiform clouds during southern West African Monsoon Season, https://doi.org/10.5194/acp-2020-602

- Pedruzo-Bagazgoitia, X., de Roode, S. R., Adler, B., Babić, K., Dione, C., Kalthoff, N., Lohou, F., Lothon, M., and Vilà-Guerau de Arellano, J., 2020: The diurnal stratocumulus-to-cumulus transition over land, Atmos. Chem. Phys., https://doi.org/10.5194/acp-2019-659.

*This has been noted. We thank the reviewer for this suggestion. In the revised manuscript, we will cite these works in this paragraph.*

(3) P2-P3 : I do not understand the link between the last sentence of page 2 and the first one of page 3. Could the authors be more explicit? Or perhaps the word « consequently » is not appropriate in this sentence?

*Thank you. There was an error in these sentences. They will now read as:*

*In the night, the nocturnal LCCs have no influence on surface solar irradiance due to the absence of sunlight. However, they persist long into the morning and early afternoon hours, thus, directly influencing the amount of incoming solar irradiance. The conditions associated with these low clouds during the daytime are less documented.*

(4) P3, Line 3 : « ...have been limited to the WAM season... »

*Thanks. This will be corrected in the revised manuscript.*

(5) P3, line 70 : « Few studies which were done with simulations, reanalysis, and satellite data in the region (e.g., Adler et al., 2017; Knippertz et al., 2011; van de Linden et al., 2015; Schuster et al., 2013) have nevertheless shown results similar to those of ground observational studies to some extent. » Could the authors precise to what observational studies they referred to?

*Some findings of all these studies regarding conditions for nocturnal LLCs are similar to findings in the recent DACCIWA studies (e.g., Babic et al., 2019; Adler et al., 2019).*
*These observational studies will be referenced in the sentence.*

(6) P4, line 100-102: Could the author reword this sentence? Remove this part perhaps "and their surface heat fluxes explore » ?

*Thank you. This sentence will be rephrased. In the revised manuscript, it will read as:*

*"Other ERA5 variables are analyzed to show some of the atmospheric and surface conditions during the occurrence of LLCs in order to understand the possible interactions between the surface and the lower levels of the atmosphere."*

(7) P5, L127-134: This paragraph about upper clouds is very important at several points of view. Because the threshold on the base height (< 2 km) can mix a lot of cloud types (Stratus, Stratocumulus, Cumulus, MCS) I wonder if it would be interesting to add a statistic on the LLC top heights just to be aware of the cloud types mixed in the so large family of LLC.

*We thank the reviewer for this comment and yes, it would be interesting to show this. However, there is no cloud top heights variable in the ERA5 product.*

(8) P5, equ. 1:
- It seems to me that equation 1 gives the Moisture Flux. The moisture advection would be defined with the horizontal gradient of specific humidity. Qavd is named differently along the article: horizontal moisture flux advection, moisture advection,…

  Yes, you are right. The quantity computed is actually moisture flux (which is also known as the water vapour transport) and not the moisture advection. In the revised manuscript, we will discuss the moisture flux rather than the advection as we did not compute that. We will also change the symbol to $Q_{flux}$ as $Q_{adv}$ may suggest advection.

- The sum should be between 1 and N and not between i and N.

  Thank you. This error will be corrected in the revised manuscript.

- Why the chosen level for the moisture flux estimate is 950 hPa? Could the authors justify this choice? Why the integrated value over the vertical is not used in this study?

  We used 950 hPa level because during the rainy season the SW monsoon flow is clearly shown at this level (*please also see from one of the DACCIWA studies – Babic et al. 2019 in figure 11 showed this level*). We did not use the integrated value over the first 2 km because we plot also the winds over the moisture flux. The winds near the surface are different from winds at around 800 hPa which is also within the 2 km. Taking the average thus may not show the SW monsoon flow well.

(9) P5, L: Ek and Holstag study focuses on ABL clouds.
- The present study does not take into account all ABL clouds since a large part of them are higher than 2 km when the ABL is higher, which is very often the case in Sahelian region.
- Some Stratus or Stratocumulus low cloud are decoupled from the surface and are then not influenced by the surface flux (see previous suggested papers of Zouzoua et al., Pedruzo et al, and Lohou et al.).
  These two points should be addressed in some way.

  Thank you for these comments. The following sentences will be incorporated in the methods section:

  *Due to the definition of LLCs used, the present study does not consider the impact of the atmospheric boundary layer (ABL) that may considerably modify the altitude of the low-level cloud. Indeed, ABL clouds have their bases higher than 2km especially in the Sahel region during the dry season. Again, some studies (e.g., Lohou et al. (2020), Pedruzo et al. (2020), Zouzoua et al. (2020)) have shown that some LLCs can be decoupled from the surface and are therefore not influenced by the surface heat fluxes.*

(10) P6, Equation 2 & 3:
- CRE has two different definitions through equation 2 & 3. The authors could remove equation 2 and just keep equation 3.

  We thank the author for this suggestion. Equation 2 will be removed in the revised manuscript.

- What are the clear sky events? Are they cases with no liquid water at all in ERA5? If so, how many cases are used for CRE estimate? Or are they theoretically calculated? This is important to understand the CRE for no-LLC cases.

Clear sky events are those events when the cloud fraction in the first 2km is zero – in the ERA5 data cloud fraction can be higher than zero when liquid water is zero as also shown in referee RC3 major comment #1 and our response to the comment. There may be higher level clouds which may have led to the CRE values during No LLC events. In the revised manuscript, we will modify Table 2 (*which was not correct anyway*) to show the number of cases that were used to calculate the CRE.

(11) P6, L163: "... SW↓ is the downwelling shortwave radiation in all-sky conditions..." If I understand correctly SW↓ is rather the downwelling short wave radiation for each LLC classes instead of "all sky conditions". This could be written in first place?

$SW^{\downarrow}$ refers to the actual radiation received with all atmospheric conditions of temperature, humidity, aerosols, clouds etc. (this is why we referred to it as 'all-sky radiation'). $SW^{\downarrow}_{CS}$ on the other hand, is computed based on the conditions but assuming there are no clouds. So, for each LLC class event, there is a corresponding $SW^{\downarrow}$ value. This point will be made clearer in the revised manuscript.

(12) P6, L169: The authors should make clear in the text and the legend that this daytime distribution gathers all the seasons.

Well noted. Thank you for this comment.
The following sentence will be added to the text: *"The distribution presents the total occurrence frequency of all seasons."*

Additionally, the following will be added to the caption: *"All seasons are included in the computed frequencies."*

(13) P6, L180: "Additionally, the early morning peak in the events of LLC Class-2 could also be partly linked to contributions from tropical oceanic low-level convection which is maximum during the early morning (Yang and Slingo, 2001)." Could the authors check the Yang and Slingo paper? It seems to me that Yang and Slingo mentioned the "Tropical oceanic deep convection" (for example page 798-799 in Yang and Slingo, 2001). If this is so, the paragraph the authors certainly referred to finishes this way : "Some of these convective systems, under optimal environmental conditions, continue to grow and reach their mature stage some time later during the night and early morning". It is rather deep convection and MCS that Yang and Slingo are talking about. How do we know that LLC class 2 are deep convection if all seasons are mixed in fig 3? What is the proportion of deep-convection versus stratocumulus? In what extent the MCS can impact the statistic presented in Fig 3 for Guinean region? Did the authors try a diurnal cycle for each season?

We agree with the reviewer that Yang and Slingo were referring to deep convection and MCS. However, with the ERA5 data and based on our definition of LLC, the lower parts of these MCSs are also considered as LLCs. They showed that some of the tropical oceanic MCSs are maximum in the night and morning. The proximity of the Guinea region to the ocean means that some of these systems may move into the Guinean region. In addition, there are residual nocturnal stratus clouds in the morning (as shown in the DACCIWA studies). All these clouds together will likely lead to higher cloud fractions (LLC Class 2).

In our statement, we did not say that all Class 2 LLCs are due to the deep MCSs shown by Yang and Slingo. We indicated that those oceanic deep convections can **partly** explain the morning peak of the Class 2. However, it not possible to make the distinction between St and MCS with

the ERA5 dataset. It could be very interesting to also show the contributions of each low cloud type to the morning peak of LLC Class 2 but this cannot be achieved with the ERA5 data. We will mention this in our conclusion as a point to be considered in future studies.

Finally, yes, we have checked the diurnal cycle for each season. Please find below the diurnal cycle of occurrence frequency for each season below in Figures 3 and 4 for DJF and JAS respectively. Most of the LLC Class 2 are found during the JAS period – which are likely due to the occurrence of both stratus and MCSs.

[Figure]

*Figure 3: Occurrence frequency of the different classes in DJF*

[Figure]

*Figure 4: Occurrence frequency of the different classes in JAS*

(14) P7, All comments on Figure 3: I don't know what the authors call "well-marked diurnal cycle" or "weak diurnal cycle"? For example, 10% of 2500 cases is a larger variation (LLC class-1 Sahelian region) than 4% of 200 cases (LLC class-2 sahelian region).

We understand and agree with the reviewer's point that there is a larger variation between the lowest and highest occurrence frequencies in Class-1 than in Class-2 over the Sahel. However, the diurnal cycle for Class-2 can easily be seen because the Class 2 events are limited to only JAS while Class-1 have a much larger spread over the different months (as shown on figure 4 in the original manuscript).

(15) P7, L189-192: I agree with this comment on the type of cloud included in the statistic. That helps to understand what really means this composite diurnal cycle.

Thank you.

(16) P7, L193-214:

- I think these two paragraphs should be moved at the beginning of this section. It would be interesting for the reader to have a first insight of the seasonal variation before the diurnal one.

Thank you. We agree. We will present the seasonal variation before the diurnal variations in the revised manuscript.

- The threshold of 2 km for the cloud base might impact the Sahelian region distributions. The number of no LLC class would decrease with a higher cloud base height threshold whereas LLC class-2 number would increase. That means that MCS, for example, can be sorted out as no-LLC class when ABL is higher than 2 km. Is that possible? In what extent this 2-km threshold impacts the atmospheric conditions for the different classes?

Yes, you are right. With a higher threshold No LCC events may decrease. We will mention this important point in the revised manuscript. LLC Class-2 may not necessarily increase (but rather all LLCs) because it will depend on the cloud fraction at the base of such MCSs. In our opinion, having a higher than 2km threshold may lead to significant changes in the analysis of the surface heat fluxes but not the moisture flux in the lower atmosphere.

(17) P8, L228, Table 2:

- The total number of hours in season indicated in Table 2 should be presented and commented in the legend and the text?

Thank you. We will comment the total number of hours in the text and in the caption of the figure.

- There is zero No-LLC events detected in Guinean region during DJF season according to table 2. But there are atmospheric conditions for this class in figures 6 and 11 for example. I guess the table is wrong.

You are right. The values presented in the table were not correct. This has been corrected and will be shown in the revised manuscript. Thank you for bringing this to our attention.

(18) P8, L231: "...cold moist air from the ocean associated with the strong southwesterly winds (positive)..." Figure 5 and 6 are very nice. I am not used in analysing the moisture flux but according to the Figures 5 and 6, $Q_{adv}$ seems rather dominated by the wind intensity than the

moisture (and even less the temperature of course) from my point of view. I think, and some AMMA and DACCIWA studies show this, that the moisture advection is null in the Guinean region during the WAM (e.g. Adler et al., Babic et al.), whereas it is important in the Sahelian region. It depends on the moisture gradient. So what does the moisture flux show? The monsoon or the Harmattan horizontal expansion according to the season?

We thank the reviewer for this comment and question.

Firstly, even though the winds are strong, the moisture content is also very significant in Figures 5 and 6 in the manuscript. To show this, we show a plot of only the specific humidity in Figure 5 below (this is the equivalent of Fig 6 in the manuscript but only the moisture is plotted in color). As clearly shown in the plot, there is a significant amount of moisture in the considered windows during cloudy events and almost no moisture during no LLC events.

So, we argue that even though horizontal moisture advection may not be very important to trigger LLCs (especially in the Guinean region), the presence of moisture (the moisture flux) is important for the occurrence and maintenance of these clouds. In general, saturation can be reached in two ways; evaporation (the presence of moisture may increase evaporation) and condensation (cooling induced by the maritime cold air advection as shown in the DACCIWA studies). Therefore, the presence of moisture is also important for LLC occurrence and maintenance. Indeed, in one of the DACCIWA papers (Babic et al., 2019), they showed that the most distinct difference between stratus and stratus free events is evident in the specific humidity (in their Figure 6c). We acknowledge our discussions did not mention all of these. We will revise the discussion of the results thoroughly.

[Figure]

Figure 5: Mean specific humidity (g kg$^{-1}$) during occurrence of LLC classes in DJF.

(19) P8, L232: "Predictably, the horizontal advection of moist air is stronger during the occurrence of LLC Class-2 events than LLC Class-1 events." Moisture flux?

*Yes, this is supposed to be moisture flux and not the advection. It will be changed in the revision of the discussion of results. Thank you.*

(20) P8, L234,„ This inland advection of moist air from the ocean has been found to play a major role in cooling (Adler et al., 2019b) which in turn enhances saturation of water vapour and consequently LLC formation (Adler et al., 2019, 2017; Babić et al., 2019b)." Moist air advection cannot induce a cooling. Adler et al., Babic et al. and Lohou et al. showed that there is no moist air advection but cold air advection. It is the cold air advection by the low level jet and Atlantic inflow which induces the saturation and the cloud formation in the Guinean region. That means that the monsoon flow is not moister than the air in the Guinean region where rain events are frequent. This can be different for the Sahelian Region which is much drier than the Guinean one. Please consider to change this comment and make different comments for the two regions.

*Thank you. We agree. The whole discussion of the results on atmospheric conditions will be thoroughly revised to focus on moisture flux rather than moisture advection. We will also use the findings of the DACCIWA studies to support our discussion wherever appropriate.*

(21) P8-9, L245-259: this paragraph deals with the vertical velocity and the divergence of the horizontal wind.

- I would show only one of these parameters since they should be proportional. No need to comment both.

  *We will show only the divergence in the revised manuscript. Thank you.*

- The authors precise at the beginning that a negative value of these two parameters is favourable to cloud formation. Stratus and stratocumulus are characterized by positive vertical velocity (hPa $s^{-1}$). So this comment cannot be general and this shows the limit of searching for atmospheric conditions of a class which may mix different clouds.

  *We are grateful to the reviewer for this important point. As indicated in the response to main comment #2, this discussion will be revised.*

- "It is also important to note that during JAS, the average vertical profiles of these processes (divergence and vertical velocity) are not similar to DJF (not shown).". If this is important, perhaps the authors should comment a little more on this or show the figures for JAS.

  *Thanks. The figure will be introduced and commented as indicated in the response to main comment #2.*

- I am not sure what the reader can conclude from this analysis.

  *As mentioned in our response to your main comment #2, we could use both DJF and JAS plots of divergence/vertical velocity to postulate the type of LLC which dominates during which season, since it is not possible to explicitly make such a distinction with the ERA5 dataset.*

(22) P9-10, section 4.2:
- Could the author precise how the monthly anomaly is computed because I am very surprised by these results?

Since we used hourly data, the anomalies were first computed at hourly timescale and then the monthly means were calculated from those. Please the computation is explained below:

There are ten different values for each hour of a given day and month (10 because ten years of data is used). For example, there will be ten different heat flux values at 12UTC on 1$^{st}$ Jan, one value for each year. The mean of all these ten values are first determined. Let's call this mean value $\overline{HF}$ and call each of the individual heat flux values as $HF_i$. So, the anomaly for the heat flux value at 12UTC on 1$^{st}$ January 2006 is computed as:

$$HF_{anom\_i} = HF_i - \overline{HF}$$

This is done for all hours on all days and months for the period of the study. The monthly mean anomaly which was shown in the manuscript is then computed from the hourly anomalies.

We will explain the computation of anomalies in the revised manuscript.

- The reduction of the net radiation at surface by the clouds induces a reduction of the surface sensible and latent heat flux. So, from my point of view, the figure 8 should rather show the cloud impact on the surface energy budget than the effect of the convective flux on the clouds.

We fully agree with the reviewer. The discussion on heat fluxes presented in the previous manuscript was flawed since we are not considering heat fluxes before the clouds are formed but rather during the cloud occurrence. We are going to present a new discussion in the revised manuscript. Here is how the new discussion on heat fluxes will be:

(Please note that the wording may change in the revised manuscript. Please also note that we will introduce another figure for the fluxes themselves and use it together with the anomalies. The mean fluxes are shown below in figure 6).

*Sensible heat:*

*Sensible heat during No LLC events is higher than when there are LLCs. There are no LLCs to reflect part of the incoming shortwave radiation. As a result, the gets more heated and becomes warmer than the air in contact with it. Therefore, sensible heat is transferred from the surface to the air. This why we have large negative sensible heat as well as large anomalies (upwards fluxes are negative in ECMWF's convention).*

*During LLC occurrence, parts of the incoming SW radiation is blocked, leading to reduced warming of the surface and thus reduce transfer of sensible heat upwards. This is evidently clear, during the JAS period in both Guinea and Sahel when the mean sensible heat flux is reduced. In terms of anomalies, the Class 2 LLCs presents positive anomalies over both regions which suggests a net downward transfer of sensible heat. This could be due to the large amounts of energy released during the formation of this clouds, which tends to heat the surrounding air of the cloud – since the air is heat, it may be warmer than the surface and this will lead to transfer of heat towards the surface. On the other hand, Class 1 LLC (especially in Guinea) presents negative anomalies of sensible heat which suggests a net upwards transfer of sensible heat. As these class does not cover a large portion of the sky, the sun will still be able to heat the surface. The observation needs to be further investigated as Guinea and Sahel presents different anomalies.*

*Latent heat:*

*Latent heat is lower during events of No LLCs than when there are LLCs because not enough moisture is evaporated (due to lack of moisture, please see figure 6 in the previous manuscript and figure 4 in this document). During LLC occurrence, there are large mean negative latent heat indicating more evaporation from the surface. Interestingly, the transfer of latent heat upwards is higher in Class 1 than in Class 2. Again, in the Class 1 events, the whole sky is not covered so the surface is still being warmed and higher amounts of moisture is still being evaporated. This is also why we have mostly negative anomalies of latent heat for Class 1 in both regions, indicating water vapour transfer upwards. On the other hand, the large cloud fractions during Class 2 events means that surface heating by the sun will be reduced (although there is still some evaporation as shown the mean fluxes). This is why anomalies of latent heat are positive i.e., the energy released in the phase changes during the cloud formation leads to a higher amount of energy in the air surrounding the cloud, which are likely higher than that from the surface – hence the net downward transfer of latent heat.*

[Figure]

*Figure 6: Mean sensible and latent heat fluxes during occurrence of the different Classes.*

- This is why I do not understand how the author can find a negative anomaly of the sensible heat flux during no-LLC event.

  During No LLC events, the surface is warmer than the air in contact with it because emitted longwave radiation from the surface is not absorbed (by clouds as there are no clouds) and re-emitted into the atmosphere. Thus, sensible heat is transferred from the warmer ground upwards into the air. The ECMWF convention for vertical fluxes is negative upwards and positive downwards. Therefore, No LLC events involve larger negative sensible heat values and negative anomalies – indicating that sensible heat moves upwards from the surface.

  We will make this clearer in the revised manuscript.

- The anomalies should be discussed in comparison to the flux itself. Do the author think that an anomaly lower than 10 W m$^{-2}$ is significant when the surface flux is around 300 W m$^{-2}$ and considering the error of the surface flux in the model.

It could be significant depending on which day and hour. But we may not be able to see this since we have only shown the monthly mean anomaly. An extreme value in the individual events may significantly impact the mean monthly anomaly value. We will show the plot of the mean fluxes and make the comparison with the anomalies in the revised manuscript. Thank you for this comment.

- At last, the authors deduce from the surface flux anomaly a surface temperature anomaly. The surface heat fluxes are proportional to the temperature vertical gradient and not directly linked to the surface temperature. Such a discussion is misleading.

Thank you. Perhaps the response to third bullet point of this question and our proposed new discussion will help to improve the explanation of the fluxes and anomalies. The discussion on heat fluxes will be revised.

(23) P11, L313-316: I fully agree with this statement. The surface convective flux should be also reduced according to the cloud fraction. So how the authors can detangle the effect of the cloud on the surface flux from the effect of the flux on the cloud triggering?

Thank you for this question. As shown in the proposed new discussion of the heat fluxes, we will rather discuss the impact of the clouds on the heat fluxes and not the other way.

**References**

Adler, B., Babia, K., Kalthoff, N., Lohou, F., Lothon, M., Dione, C., Pedruzo-Bagazgoitia, X., and Andersen, H.: Nocturnal low-level clouds in the atmospheric boundary layer over southern West Africa: An observation-based analysis of conditions and processes. Atmos Chem Phys, 19, 663–681, https://doi.org/10.5194/acp-19-663-2019, 2019

Babić, K., Kalthoff, N., Adler, B., Quinting, J. F., Lohou, F., Dione, C., and Lothon, M.: What controls the formation of nocturnal low-level stratus clouds over southern West Africa during the monsoon season? Atmos Chem Phys, 19, 13489–13506, https://doi.org/10.5194/acp-19-13489-2019, 2019

---

## Author Comment (AC2) · 24 Aug 2020

**Responses to Referee 2**

Daytime low-level clouds in West Africa – occurrence, associated drivers and shortwave radiation attenuation *by Danso et al.*

On behalf of all the authors, I wish to thank the reviewer for the thorough assessment of our study and for providing us with comments to improve the manuscript during the revision. Please see below the detailed responses to each of the reviewer's comments/questions. The reviewer's comments are shown in black font while our responses are shown in blue font. Where applicable, the changes that will be made in the manuscript are shown in italics.

This paper presents the analysis of the daytime low level clouds (LLC) over West Africa using ERA5 dataset for the period 2006-2015 to investigate their occurrence frequency, seasonal and diurnal cycles, as well as the associated atmospheric conditions for the two contrasting regions, i.e. Sahelian and Guinean region. Based on the cloud fraction, three classes were formed (no LLC, LLC class-1, and LLC class-2) and the results indicate that during the summer months LLC class-1 has the peak occurrence frequency in the Sahel, while LLC class-2 is dominant in the Gulf of Guinean. Finally, this study also addressed the attenuation of the shortwave downwelling radiation due to the LLC and found that during the summer months it is on average between 44 % and 49 % for the Sahel and Guinean region.

Considering the importance of the LLC for the regional climate of West Africa and the general lack of studies addressing these clouds, I find the topic of this study to be relevant and suitable for publication in ESD. This study provides new results which in my opinion will be useful, especially considering the ongoing and future solar energy project. However, before this manuscript can be accepted for the publication, it should be carefully revised, since there are some shortcomings in the analysis and the presentation of the results needs to be more clear. My comments and suggestions are listed below.

We thank the reviewer for the overall positive and constructive comments provided to improve our study.

Major comments:

1. I do not understand why authors extract reanalysis data of a higher resolution of 0.25o to a very coarse 1o resolution. What do you mean by "directly extracted" (Pg. 3, line 87)? Although the focus is on the analysis of large scale conditions, it is well known that some mesoscale processes are responsible for the formation and maintenance of LLC. Have you performed some sort of validation test to make sure that the conditions are properly presented with the data of coarser resolution? Additionally, sensible and latent heat flux analyzed here are not large scale phenomena.

   During the retrieval of ERA5 data, there are options to select the resolution of the grid (finer or coarser than the native grid) onto which the data will be given. In our case, we chose to retrieve the data on a 1°x1° grid. This is what we meant by 'directly extracted'. Retrievals and downloads at the finer resolutions could be very slow (especially for the

data over several pressure levels). And since most of our analyses are based on large-scale features (moisture and winds), we have decided to use the 1° grid. However, we performed some comparison tests before going ahead to use the 1° resolution data.

We made a comparison of the moisture flux and winds during LLC occurrence for the 1° resolution (Figure 1 upper row) and native ERA5 resolution = 0.25° (Figure 2 lower row). This comparison was made for only a short sub-period of 2006-2015. This comparison does not present any major discrepancy. Synoptic conditions seen by using the fine resolution are also presented very well by using the coarse resolution. Here we show only one month (August 2006) and for LLC occurrence in the Guinean region.

[Figure]

*Figure 1: Comparison of moisture flux and winds for LLC occurrence using ERA5 data at 1° (top) and 0.25° (bottom) resolutions.*

With regards to latent and sensible heat fluxes, we agree that these are not large-scale features, and this should at least be mentioned in the manuscript. In the revised manuscript, we will modify the sentence that introduces the latent and sensible heat fluxes. It will read as:

*"The atmospheric variables include specific humidity, zonal and meridional wind components, and the vertical velocity. The surface fluxes to be analyzed are the sensible and latent heat fluxes, although it should be noted that these are not large-scale phenomena."*

2. In my view, the analysis of the atmospheric conditions should be presented more consistently and more clearly.

The definition of the "horizontal moisture flux advection" is not correct (Pg. 5, line 141). Namely, eq. (1) presents the average moisture flux and not the advection of the moisture! Therefore, the presentation and discussion of the results regarding the advection of the moisture need to be carefully revised. If Figs. 5 and 6. show the quantity calculated according to the eq. (1), then they do not show moisture advection. The large majority of the paper discusses the role of moisture advection, however, this quantity is not calculated.

We thank the reviewer for this important comment. We acknowledge that the moisture flux advection was not computed. We will revise our manuscript, so that the discussion of our analysis of the atmospheric conditions will be based on the moisture flux, rather than the advection.

On Pg. 6 Equation (2) is inconsistent with the text on lines 160-161: shouldn't the equation read as: CREswt=SWt-SWcs(t)?

You are absolutely right. We only presented eq2 as CREswt= SWcs(t)-SW(t) because we did not want to present CRA with negative values. To rectify this, we will define CRA in the revised manuscript, as the absolute value of SW(t)-SWcs(t). Please also note that based on the reviewer RC1 comment #10, we will remove equation 2 and just keep eqn 3. The new equation 2 will now be:

$$CRE_{SW\downarrow}(t) = \frac{\left|SW^{\downarrow}(t) - SW_{CS}^{\downarrow}(t)\right|}{SW_{CS}^{\downarrow}(t)} \times 100$$

3. Due to the definition of the LLC used in this study, i.e. all clouds with cloud base below 2 km, different cloud types are considered as LLC. This causes some confusion in understanding atmospheric forcing related to different LLC classes in the two regions. For example, the dominant LLC class-2 during the JAS period in the Guinean region is most likely related to the stratiform clouds and shallow cumulus, while the LLC class-2 peak in the same season most likely corresponds to deep convective clouds (as it corresponds to rain events). These different cloud types have different forcings and the authors need to make a more clear distinction between these throughout the manuscript. This is especially confusing in the Abstract.

The authors should refer to the recent finding from the DACCIWA regarding the physical processes responsible for the formation and maintenance of the LLC over the Gulf of Guinea during the WAM season.

We thank the reviewer for this comment. In the revised manuscript, we will make the discussion clearer regarding which cloud types are likely to be associated with each LLC class. We will also discuss the effects of having multiple cloud types in our definition on the results obtained. We will use especially the findings of Lohou et el., (2020) and Zouzoua et al., (2020) when discussing the kinds of LLCs included in our work. For example they showed that some LLC can decouple from the boundary layer into higher altitudes and are therefore not influenced by surface fluxes. Babic et al. (2019) and Adler et al. (2019) provides detailed discussions on the processes that trigger LLCs. We will support our discussions with their findings.

4. In the recent DACCIWA papers (ACP special issue available at https://www.atmoschem-phys.net/special_issue914.html), the advection of the cold maritime air mass, related to the low level jet and the SW monsoon flow, is found to be the key process responsible for the LLC formation and not the advection of moist air. On the other hand, the advection of the relatively moist air could be important for the LLC formation in the Sahel. However, the authors do not assess the role of temperature advection. What is the reason for this?

As you mentioned in comment #3, the different cloud types have different forcings, and for low-level stratiform clouds, cold air advection is very important as shown in the DACCIWA papers (e.g., Adler et al. 2019). We actually computed the horizontal temperature advection ($H_{adv\_T}$) but did not introduce the figure in the manuscript. $H_{adv\_T}$ was computed as follows:

$$H_{adv\_T} = -\left(u\frac{\partial\theta}{\partial x} + v\frac{\partial\theta}{\partial y}\right),$$

where u and v are the zonal and meridional wind components respectively, and $\theta$ is the potential temperature.

The vertical profile of the $H_{adv\_T}$ for the first 2km is shown below in Figure 2 for LLC occurrence (Class 1 and 2) for the JAS season. The profile shown here (for the first 1km) is somehow similar to some of the results from the DACCIWA papers (*HADV in Adler et al., 2019 Figure 6a*), however the values here are rather very low compared to what was is shown in Adler et al., (2019). This difference is probably due to the fact that our LLC definition combines different low clouds (all types of clouds below 2km) – stratiform clouds probably dominate during JAS and explains the cold air advection as shown in Figure 2. Again, the events analyzed during the DACCIWA campaign are much lower than those we analyzed – averaging over all these events could have led to these low values.

On the other hand, we do not expect moisture flux to differ significantly for the different low clouds. As we know, saturation can be reached in two main ways: by evaporation (evaporation may be higher when the available moisture is high – we showed high moisture for the cloudy events) and condensation (cooling – cold air advection as shown in the DACCIWA papers). For instance, in one of the DACCIWA papers (Babic et al. 2019) the most distinct difference between stratus and stratus free events was evident in the specific humidity (please see their Figure 6c) – which shows the importance of the moisture flux for the occurrence and maintenance of these low clouds. Therefore, we decided to focus our analysis on the moisture flux in the atmosphere during the occurrence of LLCs. In the revised manuscript, we will discuss the effect of including all low clouds in our LLC definition on the results obtained and we will mention that the role of forcings such as the temperature advection as shown in the DACCIWA papers, may not be clearly noticed. We will support our discussion with some of the findings of the DACCIWA studies (especially with regards to cold air advection) where applicable.

[Figure]

*Figure 2: Vertical profiles of horizontal advection of temperature for LLC occurrence in Guinea and Sahel*

Minor comments:

- Pg. 2, line 59-60: This statement is not correct. In the Guinean region the LLC form during the night and persist long into the morning and early afternoon hours, therefore have a direct impact on the surface solar irradiance. Please see the study of Lohou et al. (2020) and Zouzoua et al. (2020) regarding this.

  Thanks. The statement will be revised. It will now read as:

  *In the night, the nocturnal LCCs have no influence on surface solar irradiance due to the absence of sunlight. However, they persist long into the morning and early afternoon hours, thus, directly influencing the amount of incoming solar irradiance.*

- Pg. 7, lines 198-200: Here it would be better to refer to recent DACCIWA publications which show that the advection of cold air, and not the advection of moist air, is the key process responsible for the formation of LLC in the Guinean region during the monsoon season.

  We thank the reviewer for this comment. We will revise the sentences and include the DACCIWA references here.

- Pg. 8, line 234-236: The reference here should be Adler et al. (2019) not (2019b). Additionally, all the studies referenced here find that the advection of cold air is the major factor in leading to saturation, not moist air! The authors should carefully revise the paper when discussing the processes leading to the formation of LLC in the two regions and when referring to previous studies to avoid making mistakes like these. Namely, based on the DACCIWA observational data it became clear that the advection of cold maritime air is

the dominant process for the LLC formation in the Gulf of Guinea, while the advection of moist air into the Sahelian region is the most dominant process for LLC formation.

Noted. The manuscript will be revised so that the appropriate terms (e.g., cold air advection instead of moist air for the Guinean region) will be used in the discussion when making references to the DACCIWA observational studies.

- Pg. 8, line 242: It should be Saharan Heat Low.

Thanks. This will be corrected in the revised manuscript.

- Pg . 8, line 243: What is the role of the cold air advection?

In this sentence, we believe *"moisture advection"* and not *"cold air advection"* was mentioned. However, the appropriate term here should be *"moisture flux"* and not "moisture advection". The sentence will be revised.

- Pg. 8, line 245: Consider here replacing "moist air" with cold air.

Noted. 'moist air' will be changed to 'cold air'.

- Pg. 9, line 287: How can water vapor be cooled by moist air advection?

This was wrong. In the revised manuscript, the sentence will read now as "…further cooled by cold air advected from the Guinean Coast, leading to saturation and then LLC formation."

- Pg. 10, line 304 and 312: Which region are you referring to? Is it entire West Africa?

In line 304, the discussion was on the Sahel region. In line 312, region being referred to was the whole of West Africa. We will make those sentences clear to indicate which regions were being referred to in those two statements.

- Pg. 10, line 310: Figs. 10 and 11 also show the downwelling shortwave radiation attenuation for no LLC class.

Yes, this is because there could be higher level clouds in those cases with no LLC. We mentioned this situation in the method section (line 127 to 129 in the original manuscript).

- Pg. 11, line 329-330: The sentence "In Sahel, the mean attenuation during the occurrence of LLC Class-1 events is 16.3% *though* areas around the southern coast of WA can experience higher losses." should be rephrased since it is not clear what is the connection between the attenuation in the Sahel and the southern coast of WA.

This will be rephrased. It will now read simply as:

*In Sahel, the mean attenuation during the occurrence of LLC Class-1 events is 16.3%.*

**References:**

Adler, B., Babia, K., Kalthoff, N., Lohou, F., Lothon, M., Dione, C., Pedruzo-Bagazgoitia, X., and Andersen, H.: Nocturnal low-level clouds in the atmospheric boundary layer over southern West Africa: An observation-based analysis of conditions and processes. Atmos Chem Phys, 19, 663–681, https://doi.org/10.5194/acp-19-663-2019, 2019

Babić, K., Kalthoff, N., Adler, B., Quinting, J. F., Lohou, F., Dione, C., and Lothon, M.: What controls the formation of nocturnal low-level stratus clouds over southern West Africa during the monsoon season? Atmos Chem Phys, 19, 13489–13506, https://doi.org/10.5194/acp-19-13489-2019, 2019

---

## Author Comment (AC3) · 24 Aug 2020

Daytime low-level clouds in West Africa – occurrence, associated drivers and shortwave radiation attenuation *by Danso et al.*

On behalf of all the authors, I wish to thank the reviewer for the thorough assessment of our study and for providing us with comments to improve the manuscript during the revision. Please see below the detailed responses to each of the reviewer's comments/questions. The reviewer's comments are shown in black font while our responses are shown in blue font. Where applicable, the changes that would be made in the manuscript are shown in italics.

This manuscript investigates daytime (06-17h) low-level clouds (LLC) over West Africa based on ERA5 (2006–2015) in two regions, the Sahel and Guinea Coastal region. LLC taken from the ERA5 archive describes cloudiness below 800 hPa (~2km). Foci of the study are on diurnal cycles in the dry and wet seasons, the seasonal variation of LLC, the atmospheric conditions related to different classes of LLC and the impact of the latter on the incoming solar radiation.

While the article describes an overall interesting topic and contains some interesting material, it also has weaknesses that shall be addressed in a major revision.

We are grateful to the reviewer for his overall positive comments on our study.

1) There needs to be a proper explanation between ERA-5 cloud fraction that usually depends on the subgrid-scale cloud scheme and the liquid and ice water paths that are relevant for the radiation scheme. Cloud fraction can be larger than zero for relative humidities below 100% and zero liquid or ice water content. There is an interesting discussion on this in Hannak et al. (2017, JCLIM). Thus cloud occurrence frequency at a gridpoint can be defined by a sub-grid scale cloud fraction or a hydrometeor content > 0. There needs to be explanations/discussion of this issue (see below).

   Thank you for this comment. We will discuss this in the data presentation in our revised manuscript. Please also see the last paragraph of our response to your major comment #3 and Figure 4 in this document which confirms your comment regarding the definition cloud occurrence frequency.

2) My major concern that also needs some time in the revision relates to the degree of realism that hourly ERA5 data have in the representation of LLC. There was no evaluation of ERA5 in Danso et al. (2019), only a comparison of total cloud cover (all clouds, all levels) based on a very coarse cloud fraction partitioning of METAR reports for three stations in West Africa. From this one figure in the Suppl. Mat. it is not obvious that the CERES data set would be inferior to ERA5 – which it likely is according to the findings in Danso et al. I am very worried about that this study may show physically consistent errors of the underlying ERA5 model. For example high LLC fractions/frequencies may be related to errors in the Bown ratio in the ERA5 model etc (with causality being another point of concern). The Reviewer proposes two ways out of it: One is to use available ground and satellite observational evidence that exists, the other is to use two other re-analyses (e.g. MERRA2, JRA-55, NCEP). In terms of the former, van der Linden et al. (2015) have shown the usefulness of the 2B-GEOPROF-

LIDAR tracks for cloud occurrence frequency in mean, 250m vertically resolved profiles, including the layer below 2 km (their Figure 6). The sampling argument given in the manuscript is not robust, as is the argument of the 1x1 grid resolution needed for PV application – the purpose is to validate the usefulness of ERA5 cloudiness and for this this the combined Cloudsat-Calipso at 01:30 LT would serve the purpose. Moreover, multi-year measurements of solar incoming radiation are available from AMMA-CATCH (http://www.amma-catch.org/?lang=fr), for the Upper Ouémé site even multiyear measurements of sensible and latent heat fluxes can be obtained. Radiation measurements for Parakou and Cotonou are also available from doi: 10.6096/baobab-dacciwa.1785. Kniffka et al. (2019, ACP) have shown large errors in surface solar radiation in ERA-I and it is questionable that this has been improved a lot in ERA5. Kniffka et al. (2020, QJRMS) have also shown that short-term forecasts of weather forecasting models, among which is the ECWMF IFS model have large errors in precipitation, radiation, and cloud cover. So there are strong arguments to validate ERA5 before drawing far reaching conclusions. I prefer to use the few, but available observational evidence, but using two other reanalyses also allows inferences about the fidelity of the results. Clearly, I do not want the author to go to deep into validation, but some more validation is necessary (for some observational points and subperiods of 2006–2015).

We thank the reviewer this comment and the suggestions given.

In the revised manuscript, we will show some validations of the ERA5 data and will discuss them. We intend to show this in an appendix that will be added to our manuscript.

**Cloud cover evaluation:**

We will use the 2B-GEOPROF-LIDAR product to perform an evaluation of the ERA5 clouds in our study region. We have already started with the evaluation of ERA5 LLC data. This is explained below:

The ERA5 cloud fractions are instantaneously generated at every hour of the day. However, the instantaneous 2B-GEOPROF-LIDAR observations over the study area are not recorded exactly on the hour (e.g., in Figure 1 below, the 2B-GEOPROF-LIDAR passes over whole WA in ≈ 7 minutes from 14:08 to 14:15). In order to compare ERA5 with the satellite observations, we compute the mean cloud fraction of the two hours bounding the 2B-GEOPROF-LIDAR observation time (i.e., in this case 14 and 15UTC, figure 1c and 1d below). The closest grid points to the 2B-GEOPROF-LIDAR observation track are selected from the ERA5 dataset for this evaluation.

As already indicated, the satellite moves along its track over WA in approximately 7 minutes. This means that the time taken for the satellite to pass along its track within the Guinea or Sahel window will be much smaller. Therefore to make the evaluation of ERA5 with 2B-GEOPROF-LIDAR in each of the windows, the mean cloud fraction along the part of the satellite track passing through that window is used. The difference between the maximum and minimum longitude (latitude) along the track passing through each window is approximately 0.5° (3°).

We then compute the mean bias error (bias) and root mean square error (RMSE) between the 2B-GEOPROF-LIDAR observations and ERA5. Please note that this

evaluation is based on only data from June to December 2006. In the proposed appendix, we will perform this evaluation with a longer time series of the 2B-GEOPROF-LIDAR observations.

Please note that the overall LLC from 2B-GEOPROF-LIDAR vertical layers (from surface to 2km) was computed using the maximum-random overlap rule (Geleyn and Hollingsworth 1979) which is used by the ECMWF for computing cloud fractions.

[Figure]

*Figure 1: Top: 2B-GEOPROF-LIDAR vertical profile of cloud fraction (a, b) for a given daytime track. (a) and (b) are the same but black dashed lines delineates the locations of the Guinea and Sahel windows respectively. All cloud structures below the white dashed horizontal line, are considered as LLC in our study. Bottom: Corresponding ERA5 2D plots of LLC for the hours (c) preceding and (d) following the satellite observation time. The white line represents the satellite track for this particular observation time. (e) A comparison the latitudinal cloud fraction seen along the 2B-GEOPROF-LIDAR track and the corresponding ERA5 closest grid point.*

[Figure]

*Figure 2: ERA5 vs 2B-GEOPROF-LIDAR cloud fraction for all satellite observations from June to December 2006 in the (a) Guinea and (b) Sahel windows.*

*Table 1: Bias and RMSE of ERA5 LLC against 2B-GEOPROF-LIDAR observations. Evaluation is done for only the daytime passes of the satellite from June to December 2006*

|  | **GUINEA** | **SAHEL** |
| --- | --- | --- |
| **BIAS** | 0.072 | 0.056 |
| **RMSE** | 0.237 | 0.118 |

Although the ERA5 hourly low-level clouds data present some biases and deviations from the satellite observations as shown above, it performs reasonably well in reproducing the LLC observations.

**Radiation evaluation:**

We will also use the surface observations from the AMMA-CATCH DB to perform an evaluation of the ERA5 surface incoming solar radiation. We will also include other datasets such as SARAH2, CERES and MERRA2 in order to compare the relative performance of ERA5. Figure 3 shows the evaluation of ERA5 against the surface observation performed for one station. Details on this evaluation will be provided in the proposed appendix to be included in the revised manuscript, as well as evaluation results from other stations with data available. We will also include a Taylor's diagram for the validation of ERA5 surface radiation data.

[Figure]

*Figure 3: A plot of bias against RMSE for ERA5 (and other datasets) against surface observations at Nalohou in Benin. The period for this evaluation is 2006 to 2015.*

With regards to the latent and sensible heat fluxes, the available time series from the AMMA-CATCH stations have a lot of missing dates (percentage of missing data ranges from 45% to 100%). We feel this will make any evaluation of the ERA5 data with these observations will be unreliable.

3) LLCs and MCSs
Having been in the region many times, I can't understand why MCSs should explain a large fraction of LLCs in the Sahel, for example. There are LLCs in the "small" leading edge/in the convective part of the MCSs, but not in the stratiform part. And MCSs are relatively infrequent. LLCs in the rainy season over the Sahel occur in the morning, but dissipate in the afternoon when isolated Cu cong or CB develop. Please comment on this.

We thank the reviewer for this comment. In our manuscript, we did not generalize that all LLCs in the Sahel are likely explained by MCSs. When we said "*It is, therefore, reasonable to assume that most of these LLC Class-2 events in the Sahel are the well documented deep MCSs which are responsible for around 90 % of total rainfall in the region*" we were referring not to the Class 1 but the Class 2. The total number of the Class 2 events are very low compared to the Class 1 (with an occurrence ratio of about

13:1). In other words, the Class 2 LLCs are infrequent in the Sahel region just like the MCSs, as you mentioned.

We however, agree with your point on LLCs and MCSs and we also acknowledge that the link we made between LLC Class 2 and MCSs is perhaps not entirely "accurate". We believe this link could be best explained by the following statement which will be introduced in the revised manuscript:

"*The probability of an MCS event is therefore higher during the occurrence of LLC Class 2 events than Class 1 events.*"

We have made some plots to illustrate this point. Figures 4 and 5 below show the vertical profile of the cloud water (liquid + ice) of 12 random events of LLC Class 1 and Class 2 respectively. Based on the work of Storer et al. (2013), the definition of an MCS is a column containing cloud water $\geq 0.01$ g·kg$^{-1}$ (red dashed line) through a continuous depth of at least 8km. These plots suggest that the likelihood of having a deep system is higher in the Class 2 events than in the Class 1 events. From the 12 sub-plots, we count around 5 cases where the cloud water is equal or higher than the threshold value starting from the lower levels and extending for at least 8km upwards. Again, we can also see the point you made about having LLCs in the convective part of an MCS in the plots.

In the revised manuscript, we will clearly highlight these points (especially the higher likelihood of an MCS occurrence during Class 2 events) and add these two figures in an appendix.

We also believe Figure 4 confirms your major comment #1 in a way. For example, we see in many of the sub-plots that water content is zero in the first 2km, however, these are all cases when cloud fraction is not zero. We will also use this figure to define the cloud occurrence frequency based on the cloud fraction rather than the hydrometeor content.

We thank you for these comments.

[Figure]

*Figure 4: Vertical profiles of cloud water content (liquid + ice) during the occurrence of LLC Class-1 events in the Sahel region. The 12 subplots are randomly extracted from the Class 1 events. The red dashed line shows the threshold for deep convective clouds as defined by Storer et al., (2013).*

[Figure]

*Figure 5: Vertical profiles of cloud water content (liquid + ice) during the occurrence of LLC Class-2 events in the Sahel region. The 12 subplots are randomly extracted from the Class 2 events. The red dashed line shows the threshold for deep convective clouds as defined by Storer et al., (2013).*

4) I have not corrected all language errors and some statements are not very clear. The author should go over the manuscript meticulously in the revision to account for this deficiency.

Thank you. In the revised version, we will check for English language errors and correct them. We will also try to rephrase some sentences when possible, to make the discussion clearer.

Minor comments:

l. 29: "efficiently represent convection AND CLOUDINESS"

Thanks. The sentence will be revised to include "cloudiness".

l. 35: "escalation" is not the right wording

The sentence will be revised. It will now read as:

*"In the light of the recent increased interest in solar energy projects in WA…"*

l. 37: "remains"

Noted and corrected. Thank you.

l. 40: prefer references in a chronological order.

Well noted. This will be revised.

l. 40: "van deR Linden", please correct.

Thank you for this correction. It will be revised.

l. 40: "Farther north"

This will be corrected. Thanks.

l. 43: "persisting into the early afternoon"

This will be corrected. Thanks.

l. 43: "as shallow convective clouds"?????

This sentence will be re-written as:

*"These LLCs consist of stratified clouds, most of which are nocturnal low stratus clouds covering wide areas and persisting into the early afternoons (Babić et al., 2019a; Schuster et al., 2013), as well as shallow convective clouds."*

l. 45-47: Here a reference to Kniffka et al. (2019, ACP) is appropriate.

Kniffka et al. (2019) will be referenced in this sentence in the revised version.

Thanks.

l 50: "The majority if these studies"

Thanks. The sentence will be revised.

l. 63: "limited TO the WAM season"

The sentence will be corrected. Thank you.

l. 84: "A better reference to ERA5 is now Hersbach et al. (2020, doi: 10.1002/qj.3803)

Thank you. We will use this reference in the revised version.

l. 101 "their surface heat fluxes explore", awkward sentence

This sentence will be rephrased. In the revised manuscript, it will read as:

*"Other ERA5 variables are analyzed to show some of the atmospheric and surface conditions during the occurrence of LLCs in order to understand the possible interactions between the surface and the lower levels of the atmosphere."*

l. 147 "horizontal air divergence", omit "air", perhaps add "wind".

Thanks. This will be revised. Then sentence will now read as:

*"In addition, the horizontal divergence of wind, and anomalies of q are analyzed together with $Q_{flux}$."*

Please note that we will change the symbol of moisture flux in the previous manuscript from $Q_{adv}$ to $Q_{flux}$ because the former may appear to suggest that the moisture advection was computed whereas the computed quantity was actually the moisture flux.

l. 182 Reword "convected air"

Please the sentence will now read as:

*"The proximity of the Guinean region to the ocean means that cold air from the Gulf of Guinea could be advected inland which will, in turn, enhance LLC formation."*

l. 188-192: Doubt that MCS contribute to LLC in reality (see major comment 3)

Please see our response to major comment #3.

l. 202-204: What about the contribution of morning LLC?

Most of the morning LLC may probably be non-precipitating during that period.

l. 212: Did Mathon et al. (2002b) really refer to LLC below 2km?

Mathon et al. (2002b) did not explicitly refer to LLC below 2km. They studied organized convective systems, some of which may have their bases below 2km.

l. 232: the adverb "predictably" seems inappropriate here. Please rephrase.

Predictably will be removed. The sentence will now read simply as:

*"The horizontal advection of moist air is stronger during the occurrence of LLC Class-2 events than LLC Class-1 events."*

l. 234 "...cold moist". Sentence terminates awkward.

Please the whole sentence will be revised based on comments from the other reviewers. The sentence will now focus on the moisture flux rather than moisture advection.

l. 240-242: q anomalies transported from the Atlantic in DJF and modulation by the WA heat low? The DJF heat low is somewhere over the Central African Republic/South Sudan at this time of the year. Please clarify.

The WAHL bit of the sentence was not correct. The sentence will be revised. It will read as:

*"Nevertheless, the LLC Class-1 events during DJF (in Sahel) are related to positive q anomalies that seem to have been transported onto the continent from the North Atlantic (see*

*Fig. A1a in Appendix) by anomalous north westerly winds but this needs to be investigated further."*

Thank you.

l. 248: Very awkward explanation of divergence. Usually, the divergence of the 2-d wind field is a good approximation of mass divergence since horizontal gradients of density are small. Please rephrase.

This is actually the definition of horizontal divergence as provided by the ECMWF. Please see https://apps.ecmwf.int/codes/grib/param-db?id=155

Section 5: The liquid and ice water content is relevant for attenuation. I am pretty sure it is not the subgrid scale cloud fraction (please clarify this point here or in the data section, see major comment 1).

Thank you. As noted in major comments #1 and #2, we will mention this point in the revised manuscript.

l. 363-365: "Other..." these processes or better features are only relevant in the wet season, not the dry season. Please mention this here.

Thank you. This will be added in the revised manuscript. It will now read as:

*"Other processes such as the atmospheric waves and jets in the region (African Easterly wave, African Easterly Jet, etc.) may also contribute to the occurrence of LLCs but these not analyzed in this study. These processes are, however, only relevant in the wet season."*

l. 393 "redaction" I think this is not the right word.

Thanks. This will be changed to *"...editing of the first draft"*.

**References**

Geleyn, J.-F. and Hollingsworth, A. An economical analytical method for the computation of the interaction between scattering and line absorption of radiation. *Beitr. Phys. Atmos.*, 1979, **52,** 1–16

Storer, R. L.; Van den heever, S. C. Microphysical Processes Evident in Aerosol Forcing of Tropical Deep Convective Clouds. *J. Atmos. Sci.*, **2013**, *70* (2), 430–446. https://doi.org/10.1175/JAS-D-12-076.1

---

## Author Response (AR1)

Dear Editor,

We wish to thank you and all three anonymous referees for considering our initial submission and for giving us the opportunuty to submit a revised version of our manuscript. We are very grateful for the thoughful and constructive comments provided by the referees to improve our manuscript. Our manuscript has been revised significantly based on the comments received and we believe this version is very much improved.

A list of the major changes made in our manuscript is presented below. Addionally, the upadated responses to each of the referees' comments and suggestions are also included in this document.

**Major changes:**
- Discussion of atmospheric conditions significantly modified and presented more clearly.
  - Focused more on moisture flux (a new figure introduced (Fig. 7) to show the moisture before and after the LLC events in addition to Figures 5 and 6).
  - Removed the analysis on divergence and vertical velocity.
  - Additional information and figures added in Appendix to improve the discussions.
- Discussion on heat flux significantly modified to show impacts of LLC on the heat fluxes rather than the opposite.
- Introduced a new figure for the mean latent heat and sensible heat fluxes in addition to the anomalies.
- Removed the figure for Bowen Ratio since it can directly be inferred from the mean sensible and latent heat fluxes.
- Included a supplementary material with details on evaluation of the ERA5 cloud fraction and surface irradiance.

**Referee Comment (RC1):** This study, based on the analysis of the ERA5 hourly data from 2006 to 2015, shows the occurrence of daytime low level clouds in WA at two different latitudes (Sahelian and Guinean). It also aims at determining the atmospheric conditions for different LLC classes (based on cloud fraction) and their impact on the solar incoming radiation. This article can be improved at several points of view: motivation, methodology explanation, results interpretation.

On behalf of all the authors, I wish to thank the reviewer for the thorough assessment of our study and for providing us with comments that helped to improve the manuscript during the revision. Please see below the detailed responses to each of the reviewer's comments/questions. The reviewer's comments are shown in black font while our responses are shown in blue font. Where applicable, the changes that have been made in the manuscript are shown in italics.

**Main comments:**

(1)     The LLC events are selected in this study with their cloud base height (< 2 km). This is in accordance with the definition of low cloud. However, the authors apply this definition at two very different places in Africa and at very contrasting seasons. The boundary layer varies from few hundred meters (during monsoon season in the Guinean region) to several kilometres (during winter season in the Sahelian region). Consequently, the LLC as defined by the authors may include or not boundary layer clouds according to the region and the season. This should be at least discussed (see specific comments) since it impacts consequently the different statistics presented in this study.

We thank the reviewer for this comment. We fully agree that the clouds analyzed in our work may or may not include all boundary layer clouds due to the definition of LLCs. In the revised manuscript, we this discuss this important issue. The following sentences, for instance, have been incorporated in Section 2.2.2:

*Due to the definition of LLCs used, the present study does not consider the impact of the atmospheric boundary layer (ABL) that may considerably modify the altitude of the low cloud. Indeed, ABL clouds have their bases higher than 2 km especially in the Sahel region during the dry season. In addition, some studies (Lohou et al., 2020; Pedruzo-Bagazgoitia et al., 2020; Zouzoua et al., 2020) have shown that some LLCs can be decoupled from the surface and are therefore not influenced by the surface heat fluxes. Therefore, having a higher cloud base threshold for LLCs may alter the analysis of the surface heat fluxes.*

(2)     In addition to the previous comment, LLC with base below 2 km gather stratus, stratocumulus, cumulus and MCS (as mentioned in the paper). These clouds develop in quite different atmospheric conditions (specifically in terms of divergence). What is the interest of highlighting the atmospheric conditions of a mix of different clouds (see specific comments)?

We agree with the reviewer that different low-level clouds may develop under different atmospheric conditions. This question somehow mirrors comment #3 by Reviewer 2. Due to the nature of the ERA5 data, it is impossible to make the distinction between the different types of low-level clouds. However, while low clouds can be associated with different types of forcing (for example in terms of divergence or cold air advection for stratiform clouds as found in the DACCIWA studies), we assume that the moisture flux (which is analyzed in our work) will not differ significantly from one LLC type to the other.

We initially planned to include the divergence/vertical velocities for the different seasons and use that to postulate the kinds of low clouds that might be dominant (as mentioned in the first response to you). However, we finally decided to entirely remove the analysis of divergence and vertical velocity and focus more on the moisture associated with the LLCs. This is to avoid confusion especially since it is known (and as you mentioned in specific comments) that the different low cloud types exhibit different conditions of divergence. We think an analysis of divergence could be very interesting, but with a dataset that can make the distinction between the different low cloud types.

(3) The cloud fraction as a parameter to define different LLC classes in order to check their impact on solar radiation seems quite logical, but not to determine the atmospheric conditions of these classes.

We thank the reviewer for this comment. However, we believe these classes could be also be important for some of the atmospheric conditions. For example, we showed that the Class 2 LLC is associated with a much stronger moisture flux (and specific humidity) compared to the Class 1 (please see Figures 5, 6, and B1 in the revised manuscript).

(4) Some explanations are missing to fully understand the analyzed atmospheric conditions, specially the surface flux anomaly.

This has been noted. We are grateful to the reviewer for this comment. In the revised manuscript, we have made a significant revision of the discussion on the surface fluxes. We have also introduced the figure for the mean flux in addition to the anomalies. We also showed how the anomalies were computed. Other questions pertaining to the surface fluxes are addressed in our responses to the specific comments.

**Specific comments:**

(1) P2, line 40: Perhaps the authors could add a reference to the work of Shrage et al. who first quantified the low level cloud frequency and fraction in southern WA.

• Schrage, J. M. and Fink, A. H.: Nocturnal continental low-level stratus over tropical West Africa: observations and possible mechanisms controlling its onset, Mon. Weather Rev., 140, 1794–1809, 2012.

Thanks. Schrage and Fink (2012) has been referenced in this sentence.

(2) P2, line 48-57: Three very recent articles published in ACP (DACCIWA special issue) should be cited here since they focus on daytime phase of low level clouds. The first one estimates, with local measurements, the impact of LLC on the surface net radiation and on the convective surface fluxes which is one objective of the present study.

• Lohou, F., Kalthoff, N., Adler, B., Babic, K., Dione, C., Lothon, M., PedruzoBagazgoitia, X., and Zouzoua, M., 2020 : Conceptual model of diurnal cycle of low-level stratiform clouds over southern West Africa. Atmos. Chem. Phys., 20, 2263–2275, doi: 10.5194/acp-20-2263-2020.

• Maurin Zouzoua, Fabienne Lohou, Paul Assamoi, Marie Lothon, Véronique Yoboue, Cheikh Dione, Norbert Kalthoff, Bianca Adler, Karmen Babić, and Xabier Pedruzo-Bagazgoitia : Breakup of nocturnal low-level stratiform clouds during southern West African Monsoon Season, https://doi.org/10.5194/acp-2020-602

• Pedruzo-Bagazgoitia, X., de Roode, S. R., Adler, B., Babić, K., Dione, C., Kalthoff, N., Lohou, F., Lothon, M., and Vilà-Guerau de Arellano, J., 2020: The diurnal stratocumulus-to-cumulus transition over land, Atmos. Chem. Phys., https://doi.org/10.5194/acp-2019-659.

We thank the reviewer for this suggestion. In the revised manuscript, we have cited these works in this paragraph.

(3)   P2-P3 : I do not understand the link between the last sentence of page 2 and the first one of page 3. Could the authors be more explicit? Or perhaps the word « consequently » is not appropriate in this sentence?

*Thank you. 'Consequently' was not supposed to be there. The sentences have been revised and now read as:*

*"In the night, the nocturnal LLCs have no influence on the surface solar radiation due to the absence of sunlight. However, they persist long into the morning and early afternoon, thus, directly influencing the amount of incoming solar irradiance and having a considerable impact on the energy balance of the earth surface as shown by Lohou et al., (2020). The conditions associated with these low clouds during the daytime in the region are less documented."*

(4)   P3, Line 3 : « ...have been limited to the WAM season... »

*Thanks. This has been corrected in the revised manuscript.*

(5)   P3, line 70 : « Few studies which were done with simulations, reanalysis, and satellite data in the region (e.g., Adler et al., 2017; Knippertz et al., 2011; van de Linden et al., 2015; Schuster et al., 2013) have nevertheless shown results similar to those of ground observational studies to some extent. » Could the authors precise to what observational studies they referred to?

*Some findings of all these studies regarding conditions for nocturnal LLCs are similar to findings in the recent DACCIWA studies (e.g., Babic et al., 2019; Adler et al., 2019).*
*These observational studies have now been referenced in the sentence.*

(6)   P4, line 100-102: Could the author reword this sentence? Remove this part perhaps "and their surface heat fluxes explore » ?

*Thank you. This sentence now reads as:*

*"Other ERA5 variables are analyzed to show some of the atmospheric and surface conditions during the occurrence of LLCs in order to understand the possible interactions between the surface and the lower atmosphere."*

(7)   P5, L127-134: This paragraph about upper clouds is very important at several points of view. Because the threshold on the base height (< 2 km) can mix a lot of cloud types (Stratus, Stratocumulus, Cumulus, MCS) I wonder if it would be interesting to add a statistic on the LLC top heights just to be aware of the cloud types mixed in the so large family of LLC.

*We thank the reviewer for this comment and yes, it would be interesting to show this. However, there is no cloud top heights variable in the ERA5 product.*

(8)   P5, equ. 1:

- It seems to me that equation 1 gives the Moisture Flux. The moisture advection would be defined with the horizontal gradient of specific humidity. Qavd is named differently along the article: horizontal moisture flux advection, moisture advection,…

*Yes, you are right. The quantity computed is moisture flux and not the moisture advection. We have significantly revised the discussion (in Section 4.1) to focus on the moisture flux rather than the moisture advection as we did not compute that. We have also changed the symbol to $Q_{flux}$ as $Q_{adv}$ may suggest advection.*

- The sum should be between 1 and N and not between i and N.

  *Thank you. This was an error. It has been corrected in the revised manuscript.*

- Why the chosen level for the moisture flux estimate is 950 hPa? Could the authors justify this choice? Why the integrated value over the vertical is not used in this study?

  *We used 950 hPa level because during the rainy season the SW monsoon flow is clearly shown at this level. We did not use the integrated value over the first 2 km because we plot also the winds over the moisture flux. The winds near the surface are different from winds at around 800 hPa which is also within the 2 km. Taking the average thus may not show the SW monsoon flow well.*

(9) P5, L: Ek and Holstag study focuses on ABL clouds.
- The present study does not take into account all ABL clouds since a large part of them are higher than 2 km when the ABL is higher, which is very often the case in Sahelian region.
- Some Stratus or Stratocumulus low cloud are decoupled from the surface and are then not influenced by the surface flux (see previous suggested papers of Zouzoua et al., Pedruzo et al, and Lohou et al.).
  These two points should be addressed in some way.

  *Thank you for these comments. We fully agree. As mentioned in the main comment #1, we have discussed this important issue in the revised manuscript.*

(10) P6, Equation 2 & 3:
- CRE has two different definitions through equation 2 & 3. The authors could remove equation 2 and just keep equation 3.

  *We thank the reviewer for this suggestion. Equation 2 was removed.*

- What are the clear sky events? Are they cases with no liquid water at all in ERA5? If so, how many cases are used for CRE estimate? Or are they theoretically calculated? This is important to understand the CRE for no-LLC cases.

  *Clear sky events are those events when the cloud fraction in the first 2km is zero – in the ERA5 data cloud fraction can be higher than zero when liquid water is zero as also shown in referee RC3 major comment #1 and our response to the comment. There may be higher level clouds which may have led to the CRE values during No LLC events. In the revised manuscript, we will modify Table 2 (which was not correct anyway) to show the number of cases that were used to calculate the CRE.*

(11) P6, L163: "... SW↓ is the downwelling shortwave radiation in all-sky conditions..." If I understand correctly SW↓ is rather the downwelling short wave radiation for each LLC classes instead of "all sky conditions". This could be written in first place?

$SW^{\downarrow}$ refers to the actual radiation received with all atmospheric conditions of temperature, humidity, aerosols, clouds etc. (this is why we referred to it as 'all-sky radiation'). $SW^{\downarrow}_{CS}$ on the other hand, is computed based on the conditions but assuming there are no clouds. So, for each LLC class event, there is a corresponding $SW^{\downarrow}$ value. This point has been made clearer in the revised manuscript (in Section 2.2.3).

(12) P6, L169: The authors should make clear in the text and the legend that this daytime distribution gathers all the seasons.

Well noted. Thank you for this comment.
The following sentence will be added to the text: *"The distribution presents the total occurrence frequency of all seasons."*

Additionally, the following will be added to the caption: *"All seasons are included in the distribution shown."*

(13) P6, L180: "Additionally, the early morning peak in the events of LLC Class-2 could also be partly linked to contributions from tropical oceanic low-level convection which is maximum during the early morning (Yang and Slingo, 2001)." Could the authors check the Yang and Slingo paper? It seems to me that Yang and Slingo mentioned the "Tropical oceanic deep convection" (for example page 798-799 in Yang and Slingo, 2001). If this is so, the paragraph the authors certainly referred to finishes this way : "Some of these convective systems, under optimal environmental conditions, continue to grow and reach their mature stage some time later during the night and early morning". It is rather deep convection and MCS that Yang and Slingo are talking about. How do we know that LLC class 2 are deep convection if all seasons are mixed in fig 3? What is the proportion of deep-convection versus stratocumulus? In what extent the MCS can impact the statistic presented in Fig 3 for Guinean region? Did the authors try a diurnal cycle for each season?

We agree with the reviewer that Yang and Slingo were referring to deep convection and MCS. However, with the ERA5 data and based on our definition of LLC, the lower parts of these MCS (the convective parts) are also considered as LLCs. They showed that some of the tropical oceanic deep convections are maximum in the night and morning. The proximity of the Guinea region to the ocean means that some of these systems may move into the Guinean region. In our statement, we did not say that all Class 2 LLCs are due to the deep MCSs shown by Yang and Slingo. We indicated that those oceanic deep convections can **partly** explain the morning peak of the Class 2. In addition, there are residual nocturnal stratus clouds in the morning (as shown in the DACCIWA studies). Having all these clouds together will likely lead to higher cloud fractions (LLC Class 2).

We believe both points were addressed in the manuscript. Since it is not possible to make the distinction (of different LLCs) with ERA5, we attributed LLC Class-2 to these kinds of clouds (residuals of nocturnal St clouds and deep clouds in the morning) as they are the clouds with largest fractional coverage. We have mentioned this limitation of the inability to distinguish between the different LLCs in our conclusion.

Finally, yes, we have checked the diurnal cycle for each season. Please find below the diurnal cycle of occurrence frequency for each season below in Figures 1 and 2 for DJF and JAS respectively. Most of the LLC Class 2 are found during the JAS period – which are likely due to the occurrence of both stratus and MCSs.

[Figure]

**Figure 1: Occurrence frequency of the different classes in DJF**

[Figure]

**Figure 2: Occurrence frequency of the different classes in JAS**

(14) P7, All comments on Figure 3: I don't know what the authors call "well-marked diurnal cycle" or "weak diurnal cycle"? For example, 10% of 2500 cases is a larger variation (LLC class-1 Sahelian region) than 4% of 200 cases (LLC class-2 sahelian region).

We understand and agree with the reviewer's point that there is a larger variation between the lowest and highest occurrence frequencies in Class-1 than in Class-2 over the Sahel. However, the diurnal cycle for Class-2 can easily be seen because the Class 2 events are limited to only JAS while Class-1 have a much larger spread over the different months (as shown on figure 4 in the original manuscript).

(15) P7, L189-192: I agree with this comment on the type of cloud included in the statistic. That helps to understand what really means this composite diurnal cycle.

Thank you.

(16) P7, L193-214:

- I think these two paragraphs should be moved at the beginning of this section. It would be interesting for the reader to have a first insight of the seasonal variation before the diurnal one.

  Thank you. We agree. We have moved the seasonal variation before the diurnal variations in the revised manuscript.

- The threshold of 2 km for the cloud base might impact the Sahelian region distributions. The number of no LLC class would decrease with a higher cloud base height threshold whereas LLC class-2 number would increase. That means that MCS, for example, can be sorted out as no-LLC class when ABL is higher than 2 km. Is that possible? In what extent this 2-km threshold impacts the atmospheric conditions for the different classes?

  Yes, you are right. With a higher threshold No LCC events may decrease. LLC Class-2 may not necessarily increase (but rather all LLCs) because it will depend on the cloud fraction at the base of such MCSs. In our opinion, having a higher than 2 km threshold may lead to significant changes in the analysis of the surface heat fluxes but not the moisture within the lower atmosphere. This issue is addressed in the revised manuscript in Section 2.2.2.

(17) P8, L228, Table 2:

- The total number of hours in season indicated in Table 2 should be presented and commented in the legend and the text?

  Thank you. We have commented the total number of hours in the text (1$^{st}$ paragraph of Section 4) and in the caption of the Table 2.

- There is zero No-LLC events detected in Guinean region during DJF season according to table 2. But there are atmospheric conditions for this class in figures 6 and 11 for example. I guess the table is wrong.

  You are right. The values presented in the table were not correct. This has been corrected. Thank you for bringing this to our attention.

(18) P8, L231: "...cold moist air from the ocean associated with the strong southwesterly winds (positive)..." Figure 5 and 6 are very nice. I am not used in analysing the moisture flux but according to the Figures 5 and 6, $Q_{adv}$ seems rather dominated by the wind intensity than the moisture (and even less the temperature of course) from my point of view. I think, and some AMMA and DACCIWA studies show this, that the moisture advection is null in the Guinean region during the WAM (e.g. Adler et al., Babic et al.), whereas it is important in the Sahelian region. It depends on the moisture gradient. So what does the moisture flux show? The monsoon or the Harmattan horizontal expansion according to the season?

We thank the reviewer for this comment and question.

Firstly, even though the winds are strong, the moisture content is also very significant. Please see Figure B1 in the revised manuscript which shows a plot of only the specific humidity. As clearly shown in the Figure B1, there is a significant amount of moisture in the considered windows during cloudy events and almost no moisture during no LLC events. This is similar to a result from Babic et al. (2019) who showed that the most distinct difference between stratus and stratus free events is evident in the specific humidity (Figure 6c in Babic et al. 2019). From that result, they suggested that the "background moisture", could be an important factor in the formation of low clouds in the region. So, the moisture flux could show the monsoon flow direction (in JAS and the harmattan in DJF) but most importantly, it also shows the background moisture levels over the region, whatever the season.

We acknowledge our original discussions did not mention all of these. In the revised manuscript, we have significantly modified the discussion presented in Section 4 to take into account all of the points mentioned above.

(19) P8, L232: "Predictably, the horizontal advection of moist air is stronger during the occurrence of LLC Class-2 events than LLC Class-1 events." Moisture flux?

Yes, this was supposed to be moisture flux and not the advection. It has been changed. Thank you.

(20) P8, L234," This inland advection of moist air from the ocean has been found to play a major role in cooling (Adler et al., 2019b) which in turn enhances saturation of water vapour and consequently LLC formation (Adler et al., 2019, 2017; Babić et al., 2019b)." Moist air advection cannot induce a cooling. Adler et al., Babic et al. and Lohou et al. showed that there is no moist air advection but cold air advection. It is the cold air advection by the low level jet and Atlantic inflow which induces the saturation and the cloud formation in the Guinean region. That means that the monsoon flow is not moister than the air in the Guinean region where rain events are frequent. This can be different for the Sahelian Region which is much drier than the Guinean one. Please consider to change this comment and make different comments for the two regions.

Thank you. We agree. The whole discussion of the results on atmospheric conditions (Section 4) has been thoroughly revised to focus on moisture flux rather than moisture advection. We also used some findings of the DACCIWA studies to support our discussion wherever appropriate.

(21) P8-9, L245-259: this paragraph deals with the vertical velocity and the divergence of the horizontal wind.

We have removed the entire analysis on divergence and vertical velocity since the different LLC types have opposing signs for these parameters. This analysis was removed from the manuscript since it is not possible to make the distinction between the different cloud types with the ERA5 data. We mentioned this limitation in our conclusion.

- I would show only one of these parameters since they should be proportional. No need to comment both.

- The authors precise at the beginning that a negative value of these two parameters is favourable to cloud formation. Stratus and stratocumulus are characterized by positive vertical velocity (hPa s$^{-1}$). So this comment cannot be general and this shows the limit of searching for atmospheric conditions of a class which may mix different clouds.

- "It is also important to note that during JAS, the average vertical profiles of these processes (divergence and vertical velocity) are not similar to DJF (not shown).". If this is important, perhaps the authors should comment a little more on this or show the figures for JAS.

- I am not sure what the reader can conclude from this analysis.

(22) P9-10, section 4.2:
- Could the author precise how the monthly anomaly is computed because I am very surprised by these results?

Since we used hourly data, the anomalies were first computed at hourly timescale and then the monthly means were calculated from those. Please the computation is explained below:

There are ten different values for each hour of a given day and month (10 because ten years of data is used). For example, there will be ten different heat flux values at 12UTC on 1$^{st}$ Jan, one value for each year. The mean of all these ten values are first determined. Let's call this mean value $\overline{HF}$ and call each of the individual heat flux values as $HF_i$. So, the anomaly for the heat flux value at 12UTC on 1$^{st}$ January 2006 is computed as:

$$HF_{anom\_i} = HF_i - \overline{HF}$$

This is done for all hours on all days and months for the period of the study. The monthly mean anomaly which was shown in the manuscript is then computed from the hourly anomalies.

This has been explained in the revised manuscript (Section 4.2).

- The reduction of the net radiation at surface by the clouds induces a reduction of the surface sensible and latent heat flux. So, from my point of view, the figure 8 should rather show the cloud impact on the surface energy budget than the effect of the convective flux on the clouds.

We fully agree with the reviewer. The discussion on heat fluxes presented in the previous manuscript was flawed since we are not considering heat fluxes before the clouds are formed but rather during the cloud occurrence. We have substantially revised the discussion on heat fluxes (Section 4.2) to show the effects of LLC on the surface energy budget.

- This is why I do not understand how the author can find a negative anomaly of the sensible heat flux during no-LLC event.

During No LLC events, the surface is warmer than the air in contact with it because emitted longwave radiation from the surface is not absorbed (by clouds as there are no clouds) and re-emitted into the atmosphere due to the vertical temperature gradient. Thus, sensible heat is transferred from the warmer ground upwards into the air. The ECMWF convention for vertical fluxes is negative upwards and positive downwards. Therefore, No LLC events involve larger negative sensible heat values and negative anomalies – indicating that sensible heat moves upwards from the surface.

This has been made clearer in the revised manuscript.

- The anomalies should be discussed in comparison to the flux itself. Do the author think that an anomaly lower than 10 W $m^{-2}$ is significant when the surface flux is around 300 W $m^{-2}$ and considering the error of the surface flux in the model.

  It could be significant depending on which day and hour. But we may not be able to see this since we have only shown the monthly mean anomaly. An extreme value in the individual events may significantly impact the mean monthly anomaly value. We have included the mean fluxes in the revised manuscript (it is now Figure 8 and the anomaly is Figure 9) and discussed the two together.

  Thank you for this comment.

- At last, the authors deduce from the surface flux anomaly a surface temperature anomaly. The surface heat fluxes are proportional to the temperature vertical gradient and not directly linked to the surface temperature. Such a discussion is misleading.

  Thank you. The discussion on heat fluxes has been thoroughly revised.

(23) P11, L313-316: I fully agree with this statement. The surface convective flux should be also reduced according to the cloud fraction. So how the authors can detangle the effect of the cloud on the surface flux from the effect of the flux on the cloud triggering?

Thank you for this question. As already mentioned, we have revised the discussion on heat fluxes to focus on the impacts of LLCs on the surface heat fluxes rather than the opposite.

**Responses to Referee 2**

**Referee Comment (RC2):** This paper presents the analysis of the daytime low level clouds (LLC) over West Africa using ERA5 dataset for the period 2006-2015 to investigate their occurrence frequency, seasonal and diurnal cycles, as well as the associated atmospheric conditions for the two contrasting regions, i.e. Sahelian and Guinean region. Based on the cloud fraction, three classes were formed (no LLC, LLC class-1, and LLC class-2) and the results indicate that during the summer months LLC class-1 has the peak occurrence frequency in the Sahel, while LLC class-2 is dominant in the Gulf of Guinean. Finally, this study also addressed the attenuation of the shortwave downwelling radiation due to the LLC and found that during the summer months it is on average between 44 % and 49 % for the Sahel and Guinean region.

Considering the importance of the LLC for the regional climate of West Africa and the general lack of studies addressing these clouds, I find the topic of this study to be relevant and suitable for publication in ESD. This study provides new results which in my opinion will be useful, especially considering the ongoing and future solar energy project. However, before this manuscript can be accepted for the publication, it should be carefully revised, since there are some shortcomings in the analysis and the presentation of the results needs to be more clear. My comments and suggestions are listed below.

On behalf of all the authors, I wish to thank the reviewer for the thorough assessment of our study and for providing us with comments that helped to improve the manuscript during the revision. Please see below the detailed responses to each of the reviewer's comments/questions. The reviewer's comments are shown in black font while our responses are shown in blue font. Where applicable, the changes that will be made in the manuscript are shown in italics.

**Major comments:**

1. I do not understand why authors extract reanalysis data of a higher resolution of 0.25o to a very coarse 1o resolution. What do you mean by "directly extracted" (Pg. 3, line 87)? Although the focus is on the analysis of large scale conditions, it is well known that some mesoscale processes are responsible for the formation and maintenance of LLC. Have you performed some sort of validation test to make sure that the conditions are properly presented with the data of coarser resolution? Additionally, sensible and latent heat flux analyzed here are not large scale phenomena.

   During the retrieval of ERA5 data, there are options to select the resolution of the grid (finer or coarser than the native grid) onto which the data will be given. In our case, we chose to retrieve the data on a 1°x1° grid. This is what we meant by 'directly extracted'. Retrievals and downloads at the finer resolutions could be very slow (especially for the data over several pressure levels). And since most of our analyses are based on large-scale features (moisture and winds), we have decided to use the 1° grid. However, we performed some comparison tests before going ahead to use the 1° resolution data.

   We made a comparison of the moisture flux and winds during LLC occurrence for the 1° resolution (Figure 1 upper row) and native ERA5 resolution = 0.25° (Figure 2 lower row). This comparison was made for only a small sub-period of 2006-2015 (one month of data). This comparison does not present any major discrepancy. Synoptic conditions seen by using the fine resolution are also presented very well by using the coarse resolution. Here we show only one month (August 2006) and for LLC occurrence in the Guinean region.

[Figure]

**Figure 1: Comparison of moisture flux and winds for LLC occurrence using ERA5 data at 1° (top) and 0.25° (bottom) resolutions.**

2. In my view, the analysis of the atmospheric conditions should be presented more consistently and more clearly.

The definition of the "horizontal moisture flux advection" is not correct (Pg. 5, line 141). Namely, eq. (1) presents the average moisture flux and not the advection of the moisture! Therefore, the presentation and discussion of the results regarding the advection of the moisture need to be carefully revised. If Figs. 5 and 6. show the quantity calculated according to the eq. (1), then they do not show moisture advection. The large majority of the paper discusses the role of moisture advection, however, this quantity is not calculated.

We thank the reviewer for this important comment. We acknowledge that the moisture flux advection was not computed. In the revised manuscript, we have significantly modified Section 4 which presents the analysis of atmospheric conditions. The discussion is now focused on the mainly on moisture flux.

On Pg. 6 Equation (2) is inconsistent with the text on lines 160-161: shouldn't the equation read as: CREswt=SWt-SWcs(t)?

You are absolutely right. We only presented eq2 as CREswt= SWcs(t)-SW(t) because we did not want to present CRA with negative values. In the revised manuscript, we have defined CRA as the difference, and then we take the absolute of the mean value. Please also note that based on the reviewer RC1 comment #10, we have removed equation 2 and just kept eqn 3. The new equation 2 is:

$$CRE_{SW\downarrow} = \left| \frac{1}{N} \sum_{i=1}^{N} \frac{SW^{\downarrow}(i) - SW_{CS}^{\downarrow}(i)}{SW_{CS}^{\downarrow}(i)} \times 100 \right|$$

3. Due to the definition of the LLC used in this study, i.e. all clouds with cloud base below 2 km, different cloud types are considered as LLC. This causes some confusion in understanding atmospheric forcing related to different LLC classes in the two regions. For example, the dominant LLC class-2 during the JAS period in the Guinean region is most likely related to the stratiform clouds and shallow cumulus, while the LLC class-2 peak in the same season most likely corresponds to deep convective clouds (as it corresponds to rain events). These different cloud types have different forcings and the authors need to make a more clear distinction between these throughout the manuscript. This is especially confusing in the Abstract.

The authors should refer to the recent finding from the DACCIWA regarding the physical processes responsible for the formation and maintenance of the LLC over the Gulf of Guinea during the WAM season.

We thank the reviewer for this comment. We totally agree. In the revised manuscript, we have entirely removed the analysis of divergence and vertical velocity since different low cloud types have opposing signs for these parameters. This was also pointed out by Reviewer 1. We now focus our discussion of atmospheric conditions mainly on the moisture flux and the discussion of results have been significantly modified.

Since it is impossible to make the distinction between different low cloud types in the ERA5 dataset, we attribute the low cloud types which are likely associated to certain events in a given season (and for which region). For example, we related LLC Class-2 in the Guinean region (in JAS) to stratiform clouds and linked this to the study of Lohou et al., (2020) who found that the daytime stratus clouds in the Guinean region causes a significant influence on the surface energy budget – a result that we also found in a our work (See the revised Section 4.2). This is also mentioned in the revised abstract.

We have also discussed the effects of having multiple cloud types in our definition on the results obtained (especially for surface heat fluxes (please see the revised Section 2.2.2)). Other DACCIWA studies are also used to support our work throughout the discussions.

4. In the recent DACCIWA papers (ACP special issue available at https://www.atmoschem-phys.net/special_issue914.html), the advection of the cold maritime air mass, related to the low level jet and the SW monsoon flow, is found to be the key process responsible for the LLC formation and not the advection of moist air. On the other hand, the advection of the relatively moist air could be important for the LLC formation in the Sahel. However, the authors do not assess the role of temperature advection. What is the reason for this?

As you mentioned in comment #3, the different cloud types have different forcings, and for low-level stratiform clouds, cold air advection is very important as shown in the DACCIWA papers (e.g., Adler et al. 2019 and Babic et al. 2019). We actually computed the horizontal temperature advection ($H_{adv\_T}$) but did not introduce the figure in the manuscript. $H_{adv\_T}$ was computed as follows:

$$H_{adv\_T} = -\left(u\frac{\partial \theta}{\partial x} + v\frac{\partial \theta}{\partial y}\right),$$

where u and v are the zonal and meridional wind components respectively, and $\theta$ is the potential temperature.

The vertical profile of the $H_{adv\_T}$ for the first 2km is shown below in Figure 2 for LLC occurrence (Class 1 and 2) for the JAS season. The profile shown here (for the first 1km) is somehow similar to some of the results from the DACCIWA papers (*HADV in Adler et al., 2019 Figure 6a*), however the values here are rather very low compared to what was is

shown in Adler et al., (2019). This difference is probably due to the fact that our LLC definition combines different low clouds (all types of clouds below 2km). Therefore, their combined effect could have led to such low values (as they may have different signs).

Although the LLC types have different conditions in which they occur, we do not expect the moisture flux to differ significantly for them. Therefore, we focused our revised discussion on the moisture flux (revised Section 4.1). The specific humidity component of the moisture flux, has been suggested to play a crucial role in the formation of low clouds (in Babic et al. 2019). Babic et al. (2019) found that the most distinct difference between stratus and stratus free events was evident in the specific humidity (please see their Figure 6c). Babic et al (2019) referred to this as the *"background moisture level"*. In our revised document, we showed the moisture flux before, during and after the occurrence of the LLC event (Figure 7 in the revised manuscript) and it clearly shown that the background moisture is very important for the occurrence of the LLCs. We suggested in our revised discussion that, the water vapour associated with the moisture flux is likely cooled by the cold air advection which in turn enhances saturation and consequently LLC occurrence. To support this, we added a plot of the potential temperature before and after the LLC events (Figure B2 in Appendix B) which showed that the temperature is strongly reduced before the LLC event – which may be due to the cold air advection. This was also supported with the findings of some of the DACCIWA studies.

[Figure]

**Figure 2: Vertical profiles of horizontal advection of temperature for LLC occurrence in Guinea and Sahel**

**Minor comments:**

- Pg. 2, line 59-60: This statement is not correct. In the Guinean region the LLC form during the night and persist long into the morning and early afternoon hours, therefore have a direct impact on the surface solar irradiance. Please see the study of Lohou et al. (2020) and Zouzoua et al. (2020) regarding this.

  Thanks. The statement has been revised. It now read as:

  *"In the night, the nocturnal LLCs have no influence on the surface solar radiation due to the absence of sunlight. However, they persist long into the morning and early afternoon, thus, directly influencing the amount of incoming solar irradiance and having a considerable impact on the energy balance of the earth surface as shown by Lohou et al., (2020)."*

- Pg. 7, lines 198-200: Here it would be better to refer to recent DACCIWA publications which show that the advection of cold air, and not the advection of moist air, is the key process responsible for the formation of LLC in the Guinean region during the monsoon season.

  We thank the reviewer for this comment. We revised the sentence and referenced Adler et al. (2020) and Babic et al. (2020).

- Pg. 8, line 234-236: The reference here should be Adler et al. (2019) not (2019b). Additionally, all the studies referenced here find that the advection of cold air is the major factor in leading to saturation, not moist air! The authors should carefully revise the paper when discussing the processes leading to the formation of LLC in the two regions and when referring to previous studies to avoid making mistakes like these. Namely, based on the DACCIWA observational data it became clear that the advection of cold maritime air is the dominant process for the LLC formation in the Gulf of Guinea, while the advection of moist air into the Sahelian region is the most dominant process for LLC formation.

  Thanks. The whole discussion of atmospheric conditions has been revised significantly (in the revised Section 4.1).

- Pg. 8, line 242: It should be Saharan Heat Low.

  Thanks. The whole sentence was modified due to the significant revision made in Section 4.1.

- Pg . 8, line 243: What is the role of the cold air advection?

  In this sentence, we believe *"moisture advection"* and not *"cold air advection"* was mentioned. However, the appropriate term here should be *"moisture flux"* and not "moisture advection". Due to the revision in Section 4.1, the sentence was deleted. New comments have been made.

- Pg. 8, line 245: Consider here replacing "moist air" with cold air.

  Noted. Thanks. This was changed.

- Pg. 9, line 287: How can water vapor be cooled by moist air advection?

  This was wrong. In the revised manuscript, we mentioned that importance of cold air advection for LLC formation as shown by the DACCIWA studies. The original sentence was removed and new comments have been made.

- Pg. 10, line 304 and 312: Which region are you referring to? Is it entire West Africa?

  Line 304 to 307 in the original manuscript was related to the Sahel region. However, those lines have been removed from the revised manuscript.

  The entire West Africa was being referred to in the first five lines of Section 5. This has been clarified in the revised manuscript.

- Pg. 10, line 310: Figs. 10 and 11 also show the downwelling shortwave radiation attenuation for no LLC class.

  Yes, this is because there could be higher level clouds in those cases with no LLC. We mentioned this situation in the method section (line 127 to 129 in the original manuscript), and also shown in Figure 2.

- Pg. 11, line 329-330: The sentence "In Sahel, the mean attenuation during the occurrence of LLC Class-1 events is 16.3% *though* areas around the southern coast of WA can experience higher losses." should be rephrased since it is not clear what is the connection between the attenuation in the Sahel and the southern coast of WA.

  This has been rephrased. It now reads simply as: "*In Sahel, the mean attenuation during the occurrence of LLC Class-1 events is 16.3%.*"

   Having been in the region many times, I can't understand why MCSs should explain a large fraction of LLCs in the Sahel, for example. There are LLCs in the "small" leading edge/in the convective part of the MCSs, but not in the stratiform part. And MCSs are relatively infrequent. LLCs in the rainy season over the Sahel occur in the morning, but dissipate in the afternoon when isolated Cu cong or CB develop. Please comment on this.

   We thank the reviewer for this comment. In our manuscript, we did not generalize that all LLCs in the Sahel are likely explained by MCSs. When we said "*It is, therefore, reasonable to assume that most of these LLC Class-2 events in the Sahel are the well documented deep MCSs which are responsible for around 90 % of total rainfall in the region*" we were referring not to the Class 1 but the Class 2. The total number of the Class 2 events are very low compared to the Class 1 (with an LLC Class-1 to Class-2 occurrence ratio of about 13:1). In other words, the Class 2 LLCs are infrequent in the Sahel region just like the MCSs, as you mentioned.

   We however, agree with your point on LLCs and MCSs and we also acknowledge that the link we made between LLC Class 2 and MCSs is perhaps not entirely "accurate". The part of the paragraph discussing this point has been revised and we added a figure in Appendix A of the revised manuscript to further explain. It now reads as:

   *Therefore, there is a higher likelihood of an MCS in the LLC Class-2 events than Class-1 and it is reasonable to assume that most of the LLC Class-2 events in the Sahel are the well documented deep MCSs that are responsible for around 90 % of total rainfall in the region (Mathon et al., 2002a; Goyens et al., 2012; Vizy and Cook, 2019). Indeed, a random plot. Indeed, a check of the vertical profiles of hydrometeor content of random LLC Class-1 and Class-2 events, reveals several LLC Class-2 events which that may be regarded as MCSs (Figure A1 in Appendix A).*

4) I have not corrected all language errors and some statements are not very clear. The author should go over the manuscript meticulously in the revision to account for this deficiency.

   Thank you. In the revised version, we have thoroughly revised the manuscript especially the in discussion of results. We believe the discussion is now clearer and language errors have been corrected.

**Minor comments:**

l. 29: "efficiently represent convection AND CLOUDINESS"

Thanks. *"cloudiness"* has been added to the sentence.

l. 35: "escalation" is not the right wording

The sentence has been revised. It now reads as: *"In the light of the recent increased interest in solar energy projects in WA…"*

l. 37: "remains"

Noted and corrected. Thank you.

l. 40: prefer references in a chronological order.

Well noted. All inline references have been presented in a chronological order.

l. 40: "van deR Linden", please correct.

Thank you for this. It has been corrected.

l. 40: "Farther north"

Corrected. Thanks.

l. 43: "persisting into the early afternoon"

This has been corrected. Thanks.

l. 43: "as shallow convective clouds"?????

This sentence has been rewritten as:

*"These LLCs consist of stratified clouds, most of which are nocturnal low stratus clouds covering wide areas and persisting into the early afternoons (Babić et al., 2019a; Schuster et al., 2013), as well as shallow convective clouds."*

l. 45-47: Here a reference to Kniffka et al. (2019, ACP) is appropriate.
Kniffka et al. (2019) has been referenced in this sentence in the revised manuscript. Thanks.
l 50: "The majority if these studies"
Thanks. Corrected.
l. 63: "limited TO the WAM season"
The sentence has been corrected. Thank you.
l. 84: "A better reference to ERA5 is now Hersbach et al. (2020, doi: 10.1002/qj.3803)

Thank you. We have changed the previous reference to Hersbach et al. (2020)

l. 101 "their surface heat fluxes explore", awkward sentence

The sentence was rephrased. It now reads as:

*"Other ERA5 variables are analyzed to show some of the atmospheric and surface conditions during the occurrence of LLCs in order to understand the possible interactions between the surface and the lower atmosphere."*

l. 147 "horizontal air divergence", omit "air", perhaps add "wind".

The whole sentence was removed due to changes made in the manuscript.

l. 182 Reword "convected air"

Please the sentence now reads as:

*"The proximity of the Guinean region to the ocean means that cold air over the Gulf of Guinea could be advected inland which will, in turn, enhance LLC formation."*

l. 188-192: Doubt that MCS contribute to LLC in reality (see major comment 3)

Please see our response to major comment #3 and Appendix A in the revised manuscript.

l. 202-204: What about the contribution of morning LLC?

Most of the morning LLC may probably be non-precipitating during that period.

l. 212: Did Mathon et al. (2002b) really refer to LLC below 2km?

Mathon et al. (2002b) did not explicitly refer to LLC below 2km. They studied organized convective systems, some of which may have their bases below 2km.

l. 232: the adverb "predictably" seems inappropriate here. Please rephrase.

Thank you. The whole sentence was revised. It now reads as:

*"The moisture flux is stronger during the occurrence of LLC Class-2 events than LLC Class-1 events. This is true for LLC occurrence in both regions; however, the moisture flux is more intense for LLC occurrence in Sahel in terms of its inland penetration."*

l. 234 "...cold moist". Sentence terminates awkward.

This sentence was revised entirely. It is the same as the response to the previous comment.

l. 240-242: q anomalies transported from the Atlantic in DJF and modulation by the WA heat low? The DJF heat low is somewhere over the Central African Republic/South Sudan at this time of the year. Please clarify.

The WAHL bit of the sentence was not correct. The sentence was revised. It now reads as:

*"Additionally, the LLC Class-1 events during DJF (in Sahel) are related to positive anomalies of specific humidity that seem to have been transported onto the continent from the North Atlantic (not shown)."*

Thank you.

l. 248: Very awkward explanation of divergence. Usually, the divergence of the 2-d wind field is a good approximation of mass divergence since horizontal gradients of density are small. Please rephrase.

This is actually the definition of horizontal divergence as provided by the ECMWF. Please see https://apps.ecmwf.int/codes/grib/param-db?id=155

However, we have entirely excluded the analysis of horizontal divergence and vertical velocity in the revised manuscript. As pointed out by both Reviewer 1 and 2, the different low cloud types have different forcings especially in terms of divergence and vertical velocity. Therefore, we removed this analysis to avoid a mix-up of the cloud types for these parameters.

Section 5: The liquid and ice water content is relevant for attenuation. I am pretty sure it is not the subgrid scale cloud fraction (please clarify this point here or in the data section, see major comment 1).

Thank you for this comment. The following sentences have been included in the data section (Section 2.1) to clarify this point:

*"The occurrence frequency of clouds may be defined by the hydrometeor content or the CF value. Here, it is determined based on the value of the 2D single-level CF although, the liquid and ice water paths mostly determine the attenuation of incoming solar radiation in the atmosphere. The CF depends on the subgrid-scale cloud scheme in the ECMWF IFS and can be non-zero when the cloud water content is zero (see Appendix A)."*

l. 363-365: "Other..." these processes or better features are only relevant in the wet season, not the dry season. Please mention this here.

Thank you. This will be added in the revised manuscript. It will now read as:

*"Other processes such as the atmospheric waves and jets in the region (African Easterly wave, African Easterly Jet, etc.) may also contribute to the occurrence of LLCs but these not analyzed in this study. These processes are, however, only relevant in the wet season."*

l. 393 "redaction" I think this is not the right word.

Thanks. This has been changed to *"...editing of the first draft".*

[revised manuscript text omitted]

---

## Referee Report (RR1)

REVIEW

**Daytime low-level clouds in West Africa – occurrence, associated drivers and shortwave radiation attenuation**

*Derrick K. Danso, Sandrine Anquetin, Arona Diedhiou, Kouakou Kouadio, Arsène T. Kobéa*

I thank the authors for the corrections made in the new version of the paper and the detailed responses for the attention of the reviewers. I still have some concerns about the section 4.2 which needs some corrections.

- Figure 8, text line 287: "During NoLLC events in both regions, there is a high negative sensible heating in all months". I think this sentence is a bit misleading since in Guinean region the NO LLC cases occur only in January.

- Lines 288-289: "due to … upward". This sentence is strange at this place. Why do the authors introduce a sentence about longwave radiation emitted by the surface in between two sentences about sensible tranferts? I agree with the first part of the sentence but the second part is again misleading.

- Line 292-293: "This is due to the lack...". The latent heat flux is proportional to the gradient between surface moisture and air moisture, so the Figure of annexe B can hardly explain alone the high bowen ratio. I would say that the bowen ration (SSH/LHF) has a seasonal evolution, as expected, with high value during the dry season and higher values during the monsoon season.

- Line 304-307: "In terms of anomalies…. Therefore the atmosphere tends to warm the surface by transferring sensible downward.". The author must reconsider their comment here. Positive anomalies mean that the sensible heat flux is larger during LLC class2 than for the other class. That's all!

- Please reconsider the whole paragraph down to line 312 since again the authors analyze the Fig9 as if it was the flux value instead of the flux anomaly.

- Line 308 :"As these LLCs…." Again comment about longwave radiation?

- I wonder if the authors should not rather present the anomaly of the SSH normalized by the net radiation to get free of the very large variation of this parameter which drives the surface flux.

---

## Author Response (AR2)

Dear Editor,

We wish to thank your re-assessment of our revied manuscript. We are also grateful to the three anonymous reviewers for their comment and suggestions. We have since revised our manuscript based on few comments received.

Sincerely

Derrick Danso

**Responses to Referee #1**

**Referee Comment (RC1):** I thank the authors for the corrections made in the new version of the paper and the detailed responses for the attention of the reviewers. I still have some concerns about the section 4.2 which needs some corrections.

We thank the reviewer for the positive comments on our revised manuscript and the suggestions for improvement. Please see below the detailed responses to each of the reviewer's comments. The reviewer's comments are shown in black font while our responses are shown in red font. Where applicable, the changes that have been made in the manuscript are shown in italics.

- Figure 8, text line 287: "During NoLLC events in both regions, there is a high negative sensible heating in all months". I think this sentence is a bit misleading since in Guinean region the NO LLC cases occur only in January.

  We have revised it. It now reads as:

  *"No LLC events occur almost throughout the entire year in the Sahel (except JAS) and only during January in the Guinean region. In both areas, the occurrence of these events leads to high negative sensible heating (Fig. 8) indicating an upward transfer of sensible heat."*

- Lines 288-289: "due to … upward". This sentence is strange at this place. Why do the authors introduce a sentence about longwave radiation emitted by the surface in between two sentences about sensible tranferts? I agree with the first part of the sentence but the second part is again misleading.

  The sentence has been revised to read as:

  *"Indeed, during no LLC events, much of the incoming radiation heats the surface, making it warmer than the air directly above it, leading to an upward sensible heat flux."*

- Line 292-293: "This is due to the lack...". The latent heat flux is proportional to the gradient between surface moisture and air moisture, so the Figure of annexe B can hardly explain alone the high bowen ratio. I would say that the bowen ration (SSH/LHF) has a seasonal evolution, as expected, with high value during the dry season and higher values during the monsoon season.

  We agree. We have revised the sentence: *"This could be as a result of a low moisture gradient between the soil and the air during those events especially in the dry season when both soil and air moisture are extremely low in the region. The lack of moisture in the region inhibits the formation of LLC during these events."*

- Line 304-307: "In terms of anomalies…. Therefore the atmosphere tends to warm the surface by transferring sensible downward.". The author must reconsider their comment here. Positive anomalies mean that the sensible heat flux is larger during LLC class2 than for the other class. That's all!

  Thank you. The comment *"Therefore the atmosphere tends to warm the surface by transferring sensible downward."*, has been removed. The whole paragraph has been significantly modified.

- Please reconsider the whole paragraph down to line 312 since again the authors analyze the Fig9 as if it was the flux value instead of the flux anomaly.

  Thank you for this suggestion. We have modified the paragraph. It now reads as: "*In terms of anomalies, the LLC Class-2 events present positive values (Fig. 9a) over both regions suggesting a net downward transfer of sensible*

*heat. This could be explained by two main reasons. Firstly, a larger portion of the sky is blocked during these events leading to lower sensible heating at the surface. Additionally, large amounts of energy are released in the lower atmosphere during the formation of these clouds, which in turn leads to a higher sensible heat flux in the atmosphere than when there are no LLC or when the cloud fraction is low. On the other hand, LLC Class-1 in the Guinean region presents negative anomalies of sensible heat that suggest a net upward transfer of sensible heat during these events. As these LLCs do not cover a large portion of the sky, a large amount of shortwave radiation still warms the surface. Moreover, the energy released in the atmosphere is not as much as for LLC Class-2 events. Thus, during these events, the atmosphere has a relatively lower sensible heat flux. It is rather surprising that the Sahel region presents positive anomalies (except in JAS) for sensible heat. As suggested by Bouniol et al., (2012), such a result should not be interpreted exclusively in terms of the radiative effects of clouds but rather the coupled interactions between fluctuations in temperature, water vapor, and clouds in this region. However, this is beyond the scope of the present study.*"

- Line 308 :"As these LLCs…." Again comment about longwave radiation?

  The whole paragraph has been modified as shown above. This comment was removed. Thank you.

- I wonder if the authors should not rather present the anomaly of the SSH normalized by the net radiation to get free of the very large variation of this parameter which drives the surface flux.

We thank the reviewer for this suggestion. Although this could be interesting, it is a bit far from what we want to show in this section; the effects of the LLC on the surface flux.

**Responses to Referee #2**

**Referee Comment (RC2):** The authors have addressed all my comments sufficiently. The revised manuscript is significantly improved. The results are presented clearly and the authors adequately addressed their results in comparison with the literature. This study provides new results regarding the characteristics of daytime low-level clouds in West Africa and their impact on the incoming solar radiation, thus providing good guidance for future solar energy projects.

I recommend publication of the study in ESD.

We are grateful to the reviewer for re-assessing our manuscript and for recommending it for publication pending the few technical corrections. Please see below the responses to each of the reviewer's comments. The reviewer's comments are shown in black font while our responses are shown in red font. Where applicable, the changes that have been made in the manuscript are shown in italics.

Technical comments:
L260: The sentence is a bit awkward, needs rewording.
The sentence has been revised. It now reads as:
*"The moisture fluxes presented so far are those present when LLC events are identified. However, it is also interesting to investigate the moisture flux before and after the LLCs occurrence in order to analyse their contribution to the maintenance or triggering of these systems."*

L 303 and 310: Reference Bouniol et al. (2020) is missing in the References list.
We thank the reviewer for spotting this. The reference in both is Bouniol et al. (2012) which is already in the reference list (and not 2020). It has been corrected.

L377: References should be: Adler et al. (2019) and Babic et al. (2019a).
You are right. Corrected. Thanks.

Figure B1: Please use a different color for wind vectors, since it is difficult to discern the colors in the Gulf of Guinea area.
Thanks. The color of the wind vectors has been changed. It can now be seen clearly anywhere over the region.

**Responses to Referee #3**

**Referee Comment (RC3):** First of all, I value the great effort that the authors undertook to revise the manuscript. This is an adequate response to my concerns. I found the comparison of ERA5, MERRA2, SARAH2 and CERES incoming irradiance very interesting. I also appreciate the effort of a short assessment of the quality of ERA5 low-level cloud fraction and occurrence frequency using the 2B-GEOPROF LIDAR product.

I consider the article now suitable for publication in Earth System Dynamics pending on the consideration of the following minor points:

We thank the reviewer for the positive comments on our revised manuscript and the few suggestions for improvement. Please see below the detailed responses to each of the reviewer's comments. The reviewer's comments are shown in black font while our responses are shown in red font. Where applicable, the changes that have been made in the manuscript are shown in italics.

**Supplementary Material:**

- In the equation (line 17) you write "Ymod,i", in line 18 you write "Ypred, i" think this should be the same.

Yes, you are right. Thank you for spotting this. "Ypred, i" is now used in both cases.

- Please mention at what temporal aggregation you did the analysis – daily incoming irradiance?

The analysis was done at an hourly aggregation scale. Please this was already mentioned on lines 3 and 6-7 in the supplementary material.

- I am surprised about the high correlation coefficient between the observed irradiation and those of the products mentioned above. The only explanation I have is that you did not subtract the seasonal cycle (this would be in line with the high standard deviation in the Taylor diagram). Please remove the seasonal cycle and re-do the Taylor plots since otherwise the statement is that the products reproduce the seasonal cycle – which is trivial.

We thank the reviewer for this suggestion. We have added an additional figure which presents the evaluation (with Taylor diagrams) of the deseasonalized time series. As expected, without the seasonality, the performance of ERA5 (and MERRA2) in reproducing the observations is quite poor in terms of the correlation coefficient. SARAH2 and CERES perform quite well than ERA5. We prefer to present both figures so we put them together (please see below) as (a) with seasonality and (b) without seasonality.

[Figure]

**Main text:**

According to the Supplementary Material, ERA5 overestimates low-level cloud fraction – I think this is true even if some problems with 2B-GEOPROF LIDAR exists in the lowest kilometer – I just suggest to make a cautionary note on this when you discuss the attenuation of the incoming shortwave radiation in section 6 (conclusion).

Thank you. The following sentence has been added to the last bullet point in the conclusion:

[revised manuscript text omitted]